# SHUFFLED TRANSFORMERS FOR BLIND TRAINING

## ABSTRACT

Conventional split learning faces the challenge of preserving training data and model privacy as a part of the training is beyond the data owner's control. We tackle this problem by introducing *blind training*, i.e., training without being aware of the data or the model, realized by shuffled Transformers. This is attributed to our intriguing findings that the inputs and the model weights of the Transformer encoder blocks, the backbone of Transformer, can be shuffled without degrading the model performance. We not only have proven the shuffling invariance property in theory, but also design a privacy-preserving split learning framework following the property, with little modification to the original Transformer architecture. We carry out verification of the properties through experiments, and also show our proposed framework successfully defends privacy attacks to split learning with superiority.

## 1 INTRODUCTION

Recent years have witnessed remarkable growth in deep learning applications, as deep neural networks (DNNs) have grown deeper and larger. It poses a dilemma for the thin edge device: on one hand, it lacks the computational power to individually train the models; on the other, data privacy would be violated if it sends all data to an untrusted party, *e.g.,* the cloud, to process. A paradigm called split learning (Gupta & Raskar, 2018) emerges to be a potential solution: without sharing its raw data, the edge transmits intermediate features to the cloud while offloading partial computation.

Typically, the private inputs are transformed into intermediate features by feeding through the first few layers of the DNN. The vanilla split learning still faces privacy leakages as an adversary could infer the input from the feature (Erdogan et al., 2021; Isola et al., 2017). Hence many works have proposed to remove the sensitive information from the features, such as encryption (Lee et al., 2022), adversarial learning (Xiao et al., 2020), differential privacy (Dong et al., 2019), etc. However, these works mostly sacrifice accuracy or efficiency for privacy guarantee. More importantly, the privacy threat of the model weights trained on the cloud is left to be an open problem — the trained weights reveal the privacy of the training data (Fredrikson et al., 2015; Carlini et al., 2019; Zhang et al., 2020), and should be proprietary to the data owner, *i.e.,* the edge.

We propose a novel *blind training* framework on the Transformer (Steiner et al., 2021), a state-of-the-art DNN achieving impressive accuracy performance on a wide range of tasks. Blind training means that the cloud conducts its part of computation 'in blind' — being unaware of the data or the model it trains, yet executing valid computation to assist the edge. The framework resembles the homomorphic encryption where the edge encrypts training data with its key, and feeds to the encrypted DNN hosted in the cloud. The cloud trains the DNN in ciphertext, without knowing the input or the model. Different from the cryptographic tool, our framework is built all in plaintext, and thus avoiding the hassle of encryption.

The key is to exploit the shuffle invariance property of Transformers. We discovered that Transformers have an intriguing property that each input, being an image or a sentence, can be randomly permuted within itself, to feed through the network, yet being equivalently trained to that without permutation. Despite that the previous work (Naseer et al., 2021) has recognized Transformer is ignorant of position information without position embeddings, we non-trivially found that even with position embeddings, Transformer is shuffling-invariant, proved by theories. By regarding the permutation order as a 'key,' the edge feeds shuffled training data to the cloud which performs natural training. Another interesting property we found is that, by training on the shuffled data, we inher-

ently obtain a Transformer encoder block with shuffled weights, which only yields valid results on inputs permuted by the 'key.' Hence the Transformer is 'encrypted' to train on the shuffled data. More importantly, the shuffled model can be 'decrypted' to obtain an equivalent plain network to which normal data can be fed.

Highlights of our contributions are: we discovered the intriguing shuffle invariance property of Transformers (and other models with Transformer encoder blocks as backbone), and built a privacy-preserving split learning framework on it. The framework provides shuffling-based privacy guarantees for training data, testing data, as well as the model weights. A variety of experiments are implemented to verify the properties, and demonstrate the superior performance of our scheme in terms of accuracy, privacy and efficiency.

## 2 BACKGROUND AND RELATED WORKS

**Transformer-based models** are the state-of-the-art deep neural networks and have attracted great attention in both areas of computer vision and natural language processing. Models including transformer encoder blocks as their backbone, such as Bert (Devlin et al., 2018), ViT (Dosovitskiy et al., 2020), T2T-ViT (Yuan et al., 2021), ViTGAN (Hirose et al., 2021), BEiT (Wang et al., 2022) and CoCa (Yu et al., 2022), have been achieving exceeding performance in a great many tasks.

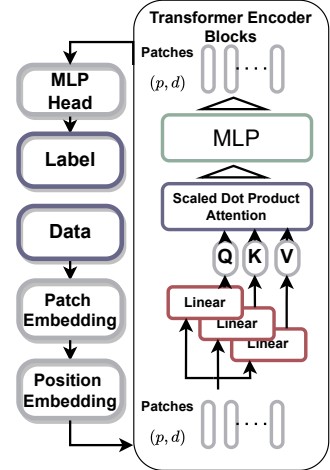

Figure 1: Transformer Encoder Block

Transformer encoder blocks, as shown in Fig. 1, mainly contain two critical components: Multi-head Scaled-dot-product self-attention and a feed-forward network (MLP). Inputs are fed in the form of patches, which are usually embedding vectors for words in Bert, or for fractions of images in ViT. The relative position of patches are learned by position embeddings (Vaswani et al., 2017), which are injected into the model. Recent studies have found that by removing the position embeddings, ViT merely loses $4\%$ accuracy on ImageNet (Russakovsky et al., 2015). And the work further reports the shuffling invariance property of ViT through experiments.

**Split learning.** As deep neural networks are growing deeper and wider, it is hardly fit for the edge which lacks the computational power but owns abundant data. Hence split learning (Gupta & Raskar, 2018) proposes to let the cloud server shoulder partial computation without accessing the data. To achieve this, a model is split into two parts, deployed on the edge and the cloud, respectively. The edge processes the first few layers and sends the intermediate features to the cloud which holds the main body of the model. If the cloud does not own the corresponding labels, it returns the prediction to the edge for computing the loss. In the backward propagation, error gradients are passed between the edge and the cloud instead of the features.

Studies have revealed that the untrusted cloud can reconstruct the private data with the intermediate features (Erdogan et al., 2021; Isola et al., 2017). Additionally, split learning allows the cloud to directly touch the model weights, which is also a threat to the privacy of training data at the edge.

**Privacy-preserving split learning.** Many efforts have been made to preserve data privacy in split learning but most have been devoted to inference data, rather than training data or model protection. Almost no lightweight protection scheme is feasible for trained model weights, which should be proprietary to the edge, and not be taken advantage of by the cloud. Traditional methods include cryptographic ones such as secure multi-party computation and homomorphic encryption. But these methods typically involve significant overhead in encryption, decryption, computation, and communication. Lee et al. (2022) implemented a polynomial approximation over nonlinear functions and encrypted the training process with FHE, but it demands 10 to 1000 times more computation power compared to plain split learning. The approximation computation also results in accuracy losses. Xiao et al. (2020) adversarially trained the edge sub-module to produce features not containing any private information, but sufficient to complete the learning task. However, the method only works when the learning converges and thus suffers potential leakage at the early stage of training. Dong

et al. (2019) inserted Gaussian noise to the features following the convention of differential privacy. Ryoo et al. (2018) adjusted the image resolution to seek a sweetspot in the tradeoff between utility and privacy. These works have to sacrifice considerable model accuracy performance to meet the privacy requirement.

**Matrix Multiplication Computation** (MMC) is a fundamental mathematical operation, and works (Lei et al., 2014; Liu et al., 2021) propose random permutation as an encryption scheme to the outsourced MMC tasks. To enhance the security, the recent work (Liu et al., 2021) introduces additive perturbation besides the multiplicative one; but their works are mostly theoretical and should be viewed as complementary to ours. We investigate the privacy guarantee in a more challenging scene — deep neural networks, and propose a practical protection scheme.

## 3 PROBLEM FORMULATION

We formally formulate our problem in the setting of split learning. The edge holds a private training data set $\mathbb{D}_{train} = \{X, Y\}$, where $X$ are the private data and $Y$ are the private labels. The edge aims at training a model with the assistance of the cloud, yet without revealing any private input or the model weights to the cloud. The cloud possesses powerful computing power but is curious about the private data of the edge and the model it trains. The edge selects a model and splits it into two parts, $F$ and $F'$, to deploy on the edge and the cloud, respectively. Referring to the loss function as $L_{task}$ and the local privacy-preserving method as $M$, the ultimate goal of the edge is to train $F$ and $F'$ jointly to

$$\underset{F,F'}{\text{minimize}}\ L_{task}(F'(F(X)), Y),\tag{1}$$

without revealing $X$ or $F'$ to the cloud or any other third party. Although the cloud does not directly access the input, it is possible to invert $X$ from $F(X)$ by the following attacks.

We assume the cloud server is honest-but-curious, meaning that it obeys the protocol and performs the learning task accordingly, but is curious about the private data. Depending on whether the edge model is accessible to the attacker, we divide the attacks into two categories:

**Black-box attacks.** The attacker is able to obtain the auxiliary data set $X_{aux}$ and the corresponding features under protection mechanism $M$ as $M(F(X_{aux}))$, which may be collected over multiple training rounds. It trains an inversion model $G$ over $(X_{aux}, F(X_{aux}))$ to invert the raw input from features. The attack goal can be

$$\underset{G}{\text{minimize}}\ L_{attack}(G(M(F^1(X_{aux})), \cdots, M^e(F^e(X_{aux}))), X_{aux}).\tag{2}$$

The superscript $e$ denotes the number of trained iterations for the features. The loss $L_{attack}$ can be the mean square error (MSE) between the reconstructed input $\tilde{X}_{aux}$ and $X_{aux}$. At convergence, $G$ works as a decoder to invert features into inputs. It should be noted that the attack we model here is different from the feature-space hijacking attack in split learning (Pasquini & Bernaschi, 2021), as the latter destroys model accuracy, inconsistent with the honest-but-curious assumption we made.

**White-box attacks.** The attacker has full access to the edge model $F$ and the protection mechanism $M$, and performs gradient descent over $M(F(X))$ and its guess $M(F(\tilde{X}))$ by

$$\underset{\tilde{X}}{\text{minimize}}\ L_{attack}(M(F(X)), M(F(\tilde{X}))).\tag{3}$$

## 4 INTRIGUING PROPERTIES OF TRANSFORMER ENCODER BLOCK

To tackle the privacy issue in split learning, it is important to perform transformations over $F(X)$ for training or inference, meanwhile preventing the adversary from inverting $X$ from $F(X)$. In our work, we adopt permutation as the transformation method, and model it by row and column shuffle, which can be expressed by matrix multiplications, i.e., given a matrix $\boldsymbol{Z} \in \mathbb{R}^{p \times d}$, the row shuffle is represented by $\boldsymbol{P}_R \boldsymbol{Z}$ where $\boldsymbol{P}_R \in \{0, 1\}^{p \times p}$ is a permutation matrix. Similarly, the column shuffle is defined as $\boldsymbol{Z}\boldsymbol{P}_C$ where permutation matrix $\boldsymbol{P}_C \in \{0, 1\}^{d \times d}$. Further explanations can be found in Appendix A. In this section, we will introduce key properties we discovered on Transformer encoder

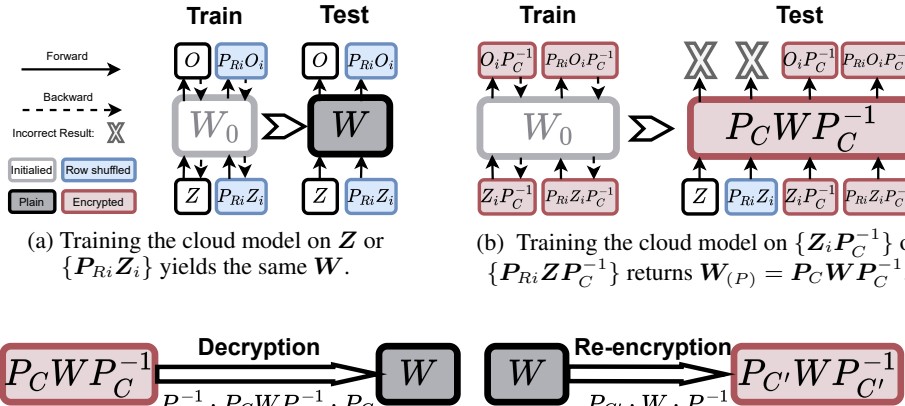

(a) Training the cloud model on $\boldsymbol{Z}$ or $\{\boldsymbol{P}_{Ri}\boldsymbol{Z}_i\}$ yields the same $\boldsymbol{W}$.

(b) Training the cloud model on $\{\boldsymbol{Z}_i\boldsymbol{P}_C^{-1}\}$ or $\{\boldsymbol{P}_{Ri}\boldsymbol{Z}\boldsymbol{P}_C^{-1}\}$ returns $\boldsymbol{W}_{(P)} = \boldsymbol{P}_C\boldsymbol{W}\boldsymbol{P}_C^{-1}$.

(c) Decryption and re-encryption of trained model can be implemented by matrix multiplication.

Figure 2: Properties of shuffled Transformers. White blocks denote initialized models, gray blocks mean the naturally trained models, and red ones indicates 'encrypted' model $\text{Enc}_{(P)}$. $\boldsymbol{O}$ represents the original outputs. $R_i$ suggests the order of row shuffle can vary for each input.

blocks, which serve the building blocks to our privacy-preserving split learning framework showed later.

Fig. 2 summarizes the properties we found and each detailed proof is provided in Appendix B. Denoting the Transformer encoder as Enc, the input matrix of Enc as $\boldsymbol{Z}$, and the row permutation matrix as $\boldsymbol{P}_R$, we have the following theorem:

**Theorem 1.** *Transformer encoder blocks are* ***row-permutation-equivalent****:*

$$Enc(\boldsymbol{P}_R\boldsymbol{Z}) = \boldsymbol{P}_R Enc(\boldsymbol{Z}) \tag{4}$$

As shown in Fig. 2a, the row permutation $\boldsymbol{P}_R$ can 'pass through' the encoder so that a reverse operation can be performed at the output of the encoder to return the original output: $\boldsymbol{P}_R^{-1}\text{Enc}(\boldsymbol{P}_R\boldsymbol{Z}) = \text{Enc}(\boldsymbol{Z})$. More interestingly, the gradients of Transformer encoder blocks in the backward propagation are invariant to row shuffle:

**Theorem 2.** *The gradients of Transformer encoder blocks w.r.t. the loss $l$ are* ***row-permutation-invariant****:*

$$\frac{\partial l}{\partial \boldsymbol{W}_{(R)}} = \frac{\partial l}{\partial \boldsymbol{W}}. \tag{5}$$

In Eq. 5, $\boldsymbol{W}$ generally refers to the weights learned naturally in the multi-head attention or the MLP of the Transformer encoder block. $\boldsymbol{W}_{(R)}$ is the corresponding weights learned on $\boldsymbol{P}_R\boldsymbol{Z}$. Hence indicating by Fig. 2a, the learned weights are the same with or without row shuffle on $\boldsymbol{Z}$. Rigorously speaking, it is not the same $\boldsymbol{W}$ learned since the weights are initialized differently; it can be considered the row shuffle is transparent to the Transformer encoder blocks, either in training or inference. Similarly, denoting $\boldsymbol{P}_C$ as the column permutation matrix, we have

**Theorem 3.** *If the Transformer encoder is permuted as $Enc_{(P)} = \boldsymbol{P}_C Enc \boldsymbol{P}_C^{-1}$, $Enc_{(P)}$ is* ***row-column-shuffle-equivalent****:*

$$Enc_{(P)}(\boldsymbol{P}_R\boldsymbol{Z}\boldsymbol{P}_C^{-1}) = \boldsymbol{P}_R Enc(\boldsymbol{Z})\boldsymbol{P}_C^{-1}. \tag{6}$$

And the outputs can be reversed by $\boldsymbol{P}_R^{-1}\text{Enc}_{(P)}(\boldsymbol{P}_R\boldsymbol{Z}\boldsymbol{P}_C^{-1})\boldsymbol{P}_C = \text{Enc}(\boldsymbol{Z})$. More intriguingly, we have

**Theorem 4.** *The gradients of Transformer encoder blocks $Enc_{(P)}$ w.r.t. the loss $l$ is* ***column-permutation-equivalent****:*

$$\frac{\partial l}{\partial \boldsymbol{W}_{(P)}} = \boldsymbol{P}_C \frac{\partial l}{\partial \boldsymbol{W}} \boldsymbol{P}_C^{-1}. \tag{7}$$

By induction, we prove that if one shuffles the normally trained Enc, it would obtain an equivalent $\text{Enc}_{(P)}$ trained on the shuffled data. Specifically, letting the weights of the normally trained Transformer encoder block be $\boldsymbol{W}$, the weights of the model trained on $\boldsymbol{P}_R \boldsymbol{Z} \boldsymbol{P}_C^{-1}$ are $\boldsymbol{P}_C \boldsymbol{W} \boldsymbol{P}_C^{-1}$, as shown in Fig. 2b. If the 'encrypted' model $\text{Enc}_{(P)}$ is deployed in the cloud, the cloud would have no clue about the model weights or the inputs, but train the model 'blindly' for the edge. Compared to cryptographic tools like fully homomorphic encryption (Lee et al., 2022), our 'encryption' method realizes a similar idea but with far less computation overhead.

Similar to the 'encryption' process, 'decryption' is feasible by reverse shuffling, as in Fig. 2c. Re-encryption is also viable by matrix multiplication, suggesting that Enc can be 'encrypted' not only by training on shuffled data, but also by permuting the trained weights.

To sum up, the row shuffle is transparent to the Transformer encoder blocks. The row-and-column-shuffle serves in a similar way to homomorphic encryption, in that the weights of $\text{Enc}_{(P)}$ are not known, while $\text{Enc}_{(P)}$ is only capable of processing shuffled $\boldsymbol{Z}$. Either by training on shuffled data or by matrix multiplication, can the weights of $\text{Enc}_{(P)}$ be obtained. The entire process does not incur additional computational burden, or sacrifice accuracy performance.

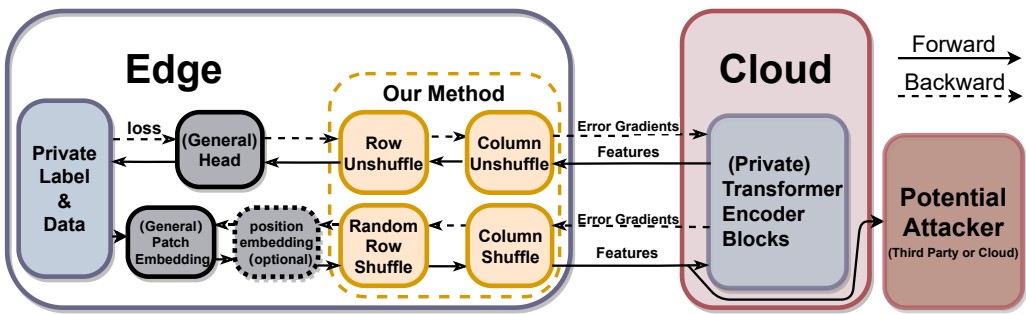

Figure 3: The structure of our privacy-preserving split learning over shuffled Transformers.

# 5 METHODOLOGY

Inspired by the permutation invariance properties of Transformer encoder blocks, we present our method to preserve training data, inference data and model weights privacy in the split learning framework, followed by the privacy indication of shuffling.

## 5.1 SHUFFLED TRANSFORMERS

The overall scheme of our method is shown in Fig. 3. We split a typical Transformer-based model into three stubs: the part from input layer to patch embeddings residing at the edge, the transformer encoder blocks at the cloud, and the MLP and loss layer at the edge. Position embedding is optional and depends on practical situations. Being transmitted between the edge and the cloud, the smashed data in the forward loop and backward loop are referred to as features and error gradients, respectively.

Our method runs at the edge, processing the smashed data transmitted back and forth between the edge and the cloud. We introduce four basic operations: row shuffle and unshuffle, column shuffle and unshuffle. All *shuffling orders are secret* of the edge. We perform row and column shuffle to the features sent from the edge to the cloud. The features $\boldsymbol{Z} = F(X)$ are expressed by a $(p, d)$ (*e.g.,* $(197, 768)$) matrix, where $p$ denotes the number of patches and $d$ the dimension of each patch. We left the batch size out of the modeling, as shuffling takes place within a single input. We shuffle the patches to be sent and unshuffle the received features by

$$\text{Shuffle: } M_{\boldsymbol{Z}}(\boldsymbol{Z}) = \boldsymbol{P}_R \boldsymbol{Z} \boldsymbol{P}_C^{-1}, \tag{8}$$

$$\text{Unshuffle: } M_{\boldsymbol{Z}}^{-1}(F'_{(P)}(M_{\boldsymbol{Z}}(\boldsymbol{Z}))) = \boldsymbol{P}_R^{-1} F'_{(P)}(M_{\boldsymbol{Z}}(\boldsymbol{Z})) \boldsymbol{P}_C, \tag{9}$$

where $\boldsymbol{P}_R \in \{0,1\}^{p \times p}$ and $\boldsymbol{P}_C \in \{0,1\}^{d \times d}$ are row and column permutation matrices, respectively. The row and column shuffle are jointly denoted as mechanism $M_{\boldsymbol{Z}}$. $\boldsymbol{P}_R$ is chosen randomly per $\dot{\boldsymbol{Z}}$,

whereas $\boldsymbol{P}_C$ is chosen per model, i.e., $\boldsymbol{P}_C$ is the same for all inputs. The model stub $F'_{(P)}$ on the cloud is trained as the usual network, and the output of Transformer encoder block is sent to the edge for unshuffle by Eq. 9. The backward loop is no different from the normal backward propagation.

**Privacy for model weights and training data.** From Thm. 4, we know that Transformer encoder blocks at the cloud trained on $M_{\boldsymbol{Z}}$ has the following unique property. Letting $F'_{(P)} = \{\boldsymbol{W}_{(P)}, b_{(P)}, \gamma_{(P)}\}$ and $F' = \{\boldsymbol{W}, b, \gamma\}$ denote the {weight matrix, bias, layer normalization parameter} of the cloud model trained on $M_{\boldsymbol{Z}}(\boldsymbol{Z})$ and the normal $\boldsymbol{Z}$, respectively, we have:

$$\boldsymbol{W}_{(P)} = \boldsymbol{P}_C \boldsymbol{W} \boldsymbol{P}_C^{-1} \triangleq M_{\boldsymbol{W}}(\boldsymbol{W}), \ \ b_{(P)} = b\boldsymbol{P}_C^{-1}, \ \ \gamma_{(P)} = \gamma \boldsymbol{P}_C^{-1}. \tag{10}$$

Note that Eq. 10 demands the weight matrix to be a square one, which requires minor modification to the cloud model as we will elaborate on later. It is an interesting fact that if the model is trained on the shuffled patches $M_{\boldsymbol{Z}}(\boldsymbol{Z})$, the randomly initialized weight $\boldsymbol{W}_0$ learns to become $\boldsymbol{W}_{(P)}$ instead of $\boldsymbol{W}$, and thus the true model weights are unknown to the cloud. Privacy for training data is preserved by $M_{\boldsymbol{Z}}$, as each input is randomly row-shuffled by $\boldsymbol{P}_R$ and column-shuffled by $\boldsymbol{P}_C$ before being sent. Note that since no particular patch size is specified, all patch size would work but the finest granularity is recommended for privacy concerns. We will give the formal guarantee in Sec. 5.2.

**Protection of the inference data.** According to Thm. 3, it is obvious that the shuffle (Eq. 8) and unshuffle (Eq. 9) procedures give legitimate testing results. Since the feature sent to the cloud is shuffled, the testing data privacy is preserved. In addition, only the entity who holds $\boldsymbol{P}_C^{-1}$ would produce valid inference results on $F'_{(P)}$:

$$F'_{(P)}(\boldsymbol{Z}) = Invalid \ Result, \ \ F'_{(P)}(M_{\boldsymbol{Z}}(\boldsymbol{Z})) = M_{\boldsymbol{Z}}(F'(\boldsymbol{Z})). \tag{11}$$

**Re-encrypting the model** is also available with our method. The edge may pre-train a secret Transformer body with a secret $\boldsymbol{P}_C$, and later can obtain the unencrypted weights $\boldsymbol{W} = \boldsymbol{P}_C^{-1} \boldsymbol{W}_{(P)} \boldsymbol{P}_C$, or re-encrypt the model to authorize other party to use it: $\boldsymbol{W}_{(P')} = \boldsymbol{P}_{C'} \boldsymbol{W} \boldsymbol{P}_{C'}^{-1}$. Other parties who hold $\boldsymbol{P}_{C'}$ can further perform privacy-preserving transfer learning or fine-tuning on the model.

**Model structure modification.** Our method requires all the weight matrices in the Transformer encoder blocks and the MLP layer to be square ones. For example, the multi-head attention in the classical ViT-Base (Dosovitskiy et al., 2020) introduces non-square weight matrices, and its MLP has an input/output dimension of 768 but a hidden layer of dimension 3072. To implement shuffled transformers, each head has to be square: weights of the linear projection layers of $Q, K, V$ are reshaped to $(768 \times no.\_of\_heads, 768)$, so that each head has a shape of $(768, 768)$. Instead of concatenating them, we calculate their average to keep the square shape of the weight matrices, which mildly increases the computational overhead but no accuracy decline. Or one can simply choose to use single-head attention to replace the multi-head one, which also has limited impact to the model performance. The MLP layer is reshaped with 768-hidden units, which keeps the weight matrices square.

## 5.2 DEFINITION OF PRIVACY

The purpose of shuffling is to prevent the attacker from reconstructing input $X$ given the feature $M_{\boldsymbol{Z}}(\boldsymbol{Z})$. In this work, we consider recovering $\boldsymbol{Z}$ to be the same with reconstructing $X$, as a black-box or white-box attacker can easily invert $X$ from $\boldsymbol{Z}$. To quantize the likelihood that an adversary rebuilds $\boldsymbol{Z}$ from $M_{\boldsymbol{Z}}(\boldsymbol{Z})$, we first define neighboring permutations as:

**Definition 1.** *(Neighboring Permutations.) For feature matrix $\boldsymbol{Z} \in \mathbb{R}^{p \times d}$, all row-column permutation orders of $\boldsymbol{Z}$ constitute $\mathbb{S}$. Any two permutations $\sigma, \sigma' \in \mathbb{S}$ are neighboring permutations.*

Our privacy definition based on shuffling is as follows.

**Definition 2.** *($\sigma$-privacy.) Given the private feature $\boldsymbol{Z}$ and permutation set $\mathbb{S}$, a randomized shuffling mechanism $M : M(\boldsymbol{Z}) \mapsto \boldsymbol{Z}' \in \mathbb{S}$ is $\sigma$-private if for all $\boldsymbol{Z}, \boldsymbol{Z}'$, and any neighbouring permutations $\sigma, \sigma'$, we have*

$$\Pr[M(\sigma(\boldsymbol{Z})) = \boldsymbol{Z}'] = \Pr[M(\sigma'(\boldsymbol{Z})) = \boldsymbol{Z}']. \tag{12}$$

$\sigma$-privacy suggests that shuffling mechanism $M$ is agnostic of the relative order of patches. Thus any adversary is incapable of telling the original order from its neighboring permutations, given the

perturbed feature $\boldsymbol{Z}'$. This definition shares some similarities with $d_\sigma$-privacy in (Meehan et al., 2021) but ours removes $\alpha$ in $d_\sigma$-privacy as permutations are sampled from a uniform distribution rather than Mallows model. With the definition, we can calculate that if all row-column permutations are equally likely for an input, our $M_{\boldsymbol{Z}}(\boldsymbol{Z})$ has the probability of $\frac{1}{p!d!}$ to reveal the true $\boldsymbol{Z}$, which is negligible. Similarly for weights shuffling, $M_{\boldsymbol{W}}(\boldsymbol{W})$ and $M_{\boldsymbol{W}}(\boldsymbol{P}\boldsymbol{W}\boldsymbol{P}^{-1})$ (where $\boldsymbol{P}$ is a random permutation matrix) have the same probability to yield $\boldsymbol{W}'$, which has $\frac{1}{d!}$ probability to be the true $\boldsymbol{W}$. Hence the mechanism prevents any adversary from recovering the true weights. Hereby we have

**Proposition 1.** *Input shuffling $M_{\boldsymbol{Z}}(\cdot)$ is $\sigma$-private with $\mathbb{S}$ being the set for all possible $\boldsymbol{P}_R, \boldsymbol{P}_C^{-1}$ permutations.*

**Proposition 2.** *Weights shuffling $M_{\boldsymbol{W}}(\cdot)$ is $\sigma$-private with $\mathbb{S}$ being the set for all possible $\boldsymbol{P}_C, \boldsymbol{P}_C^{-1}$ permutations.*

## 6 Experiments and Evaluations

We first verify the properties of the shuffled Transformer by experiments, and show its defence capability against attacks.

**Setup:** Our implementation is built on `Pytorch` and `Torchvision`. We use Cifar10 (Krizhevsky et al., 2009) consisting of 60,000 natural images in 10 classes, and CelebA (Liu et al., 2015) containing 2,022,599 faces from 10,177 celebrities. On CelebA, we adopt `timm`[1] model vit_base_patch16_224 (ViT-Base) pre-trained on ImageNet to transfer to a 40-binary-attribute classification task. We adopt a single-head ViT-Base according to the modification stated in Sec. 5.1 with the following structure: 12 layers, image size=224, patch size=16, embedding_dim=768, mlp_hidden_dim=768 and one head. The model is referred to as single-head ViT for CelebA later. A SGD optimizer is used with a cosine scheduler, for which the (initial, final) learning rate are set to $(0.05, 2 \times 10^{-4})$ and $(5 \times 10^{-4}, 2 \times 10^{-6})$ for the MLP and the encoder blocks, respectively.

On Cifar10, we use a smaller ViT with the structure: 6 layers, image size=32, patch size=4, embedding_dim=512, mlp_hidden_dim=512, 1 head in the row-column-shuffle case and 6 heads in others. The models are called the single-head and the multi-head ViT for Cifar10, respectively. These ViTs are trained from scratch on Cifar10, and thus it provides satisfying but inferior accuracy to the pre-trained one. An Adam optimizer and a cosine scheduler with learning rate $10^{-4}$ are used.

**Baselines:** We compare our method with a set of existing privacy-preserving methods. Conventional cryptographic tools are not included for unbearable computational and communication costs. Baselines include unprotected split learning (**SL**), adversarial learning (**adv**) (Xiao et al., 2020), methods based on **Transform** containing adding Gaussian noise $\sim \mathcal{N}(0, 4)$ (**GN**) (Dong et al., 2019) and **Blur** (Ryoo et al., 2018).

**Metrics:** We evaluate model performance from the accuracy, privacy, and efficiency aspects. The average classification accuracy of 40 attributes, and the 10-class classification accuracy are reported for CelebA and Cifar10, respectively. Privacy is gauged by the attackers' capability in reconstructing inputs. We select popular metrics such as Structural Similarity (SSIM), Peak Signal to Noise Ratio (PSNR) (Hore & Ziou, 2010), and F-SIM. For F-SIM, We feed the original and reconstructed inputs into a third-party network and compare the cosine similarity between the features. The third-party network for CelebA is InceptionResNetV1 of FaceNet (Schroff et al., 2015), pre-trained on VggFace2 (Cao et al., 2018), and that for Cifar10 is ResNet18 pre-trained on ImageNet.

### 6.1 Verifying Properties

We experimentally verify the properties of shuffled Transformers by their accuracy performance.

**Row-permutation-equivalence:** To verify Thm. 1 and Thm. 2, we train and test models with or without row-shuffled (RS) inputs. No model modification are needed for row shuffle and thus the original model structures are used. Testing accuracies are reported in Tab. 1. Despite being shuffled or not, the testing data reach approximately the same accuracy for each dataset, verifying the row shuffle is indeed transparent to the Transformer encoder blocks. The model trained on row-shuffled

---

[1]https://github.com/rwightman/pytorch-image-models

data has an equivalent performance to the model normally trained, verifying their training processes are equivalent. On CelebA, removing the position embedding from the model merely leads to an 0.506% accuracy drop compared to the unprotected SL (91.908%).

Table 1: Accuracies(%) on row-shuffled and row-column-shuffled data. Shuffle methods of the testing data correspond to that of the training data.

| Test \ Train | ViT Cifar10 | | ViT-Base CelebA | | multi-head ViT Cifar10 | | single-head ViT CelebA | |
|---|---|---|---|---|---|---|---|---|
| | w/ RS | w/o RS | w/ RS | w/o RS | w/ RCS | w/o RCS | w/ RCS | w/o RCS |
| w/ Shuffle | 78.35 | 78.38 | 91.40 | 91.61 | 80.76 | 10.38 | 91.30 | 79.76 |
| w/o Shuffle | 79.08 | 78.40 | 91.62 | 91.52 | 7.50 | 80.98 | 71.50 | 91.51 |

**Row-column-shuffle-equivalence:** To verify Thm. 3 and Thm. 4, we train and test on natural or row-column-shuffled (RCS) data, and report the testing accuracies in Tab. 1. Particularly for CelebA, we pre-train the single-head ViT on RCS (natrual) ImageNet data, and transfer the weights of the $20^{th}$ epoch to CelebA. 'Training with RCS' suggests both pre-training and fine-tuning are on RCS data. Note that due to the imbalanced data distribution, the average accuracy reported for the 40 attributes of CelebA is around $66 - 73\%$ for random guesses. And the benchmark of CelebA on pre-trained ImageNet is around $91\%$. It is clear that if the model is trained on normal data but test on shuffled data, or vice versa, its performance is close to random guesses. Otherwise, the shuffled Transformer almost achieves no accuracy loss.

Table 2: Accuracies(%) of decrypted/re-encrypted models.

| | Encrypted | Decrypted |
|---|---|---|
| single-head ViT Cifar10 | 79.55 | 79.01 |
| multi-head ViT Cifar10 | 80.96 | 80.67 |
| single-head ViT CelebA | 91.51 | 91.30 |

Table 3: Accuracies(%) of CelebA models w/ and w/o Position Embedding (PE).

| | w/ PE | w/o PE |
|---|---|---|
| unprotected SL | 91.91 | 91.52 |
| RS | 91.83 | 91.40 |

**Decryption and Re-encryption:** We also verify decrypting and re-encrypting the weight matrices are feasible with Eq. 10. For each model, we 'encrypt' the naturally trained model by multiplying a random permutation matrix and have it tested on the shuffled test data. And we also 'decrypt' the model trained on row-column-shuffled data and test it on plain data. All the testing accuracies are reported in Tab. 2, and the results are almost no different from the benchmarks.

**Position embeddings:** Our method has no influence on network parameters on the edge, including positional embeddings (see Appendix B.6 and C.6 for more detail). As shown in Tab. 3, shuffling has little impact to accuracy despite the position embeddings exist or not.

Further verification experiments on NLP tasks and tabular data can be found in Appendix C.3&C.4.

## 6.2 DEFENCE AGAINST ATTACKS

We focus on the privacy performance of our method against attacks in Sec. 3, in comparison to the baselines. As we observe, position embeddings hurt training data privacy in our black-box attack since such an attack is very strong, and almost does not affect accuracy, we choose to remove position embedding in training.

**Defence in inference.** The train set of CelebA is adopted as $X_{aux}$ and the test set is used as the private inference data. All defence methods are applied to the same edge model $F$, a fixed patch_embedding layer.

In inference, the black-box attacker can only acquire smashed data once, and thus we fix weights of the edge model and train the attacker model to minimize the loss of Eq. 2 where $e = 1$. We implement the attacker with an MAE decoder $G$, pre-trained on ImageNet with an additional position embedding layer at the head of the decoder and a Tanh activation layer at the rear. We report the inference accuracy and attack performance in Tab. 4. It is observed that adv and Blur fails to maintain a privacy guarantee. GN successfully prevents the attacker from reconstructing the private inputs but degrades the accuracy considerably. Our methods achieve satisfying privacy performance across all metrics while sharing a close accuracy to SL.

Table 4: Accuracy and privacy on the inference data, CelebA. ↓ means desirable direction.

| | Utility | Privacy in Black-Box | | | Privacy in White-Box | | |
|---|---|---|---|---|---|---|---|
| | Accuracy↑ | SSIM↓ | PSNR↓ | F-SIM↓ | SSIM↓ | PSNR↓ | F-SIM↓ |
| unprotected SL | 91.91% | 0.645 | 16.247 | 0.933 | 0.173 | 12.411 | 0.738 |
| adv | 91.82% | 0.539 | 12.123 | 0.767 | 0.173 | 12.411 | 0.738 |
| Blur | 89.84% | 0.501 | 14.233 | 0.501 | 0.079 | 10.729 | 0.124 |
| GN | 82.40% | 0.290 | 13.052 | 0.361 | 0.078 | 9.968 | 0.127 |
| Our RS | 91.83% | 0.243 | 10.395 | 0.299 | 0.315 | 10.519 | 0.139 |
| Our RCS | 91.58% | 0.270 | 10.629 | 0.345 | 0.160 | 10.464 | 0.107 |

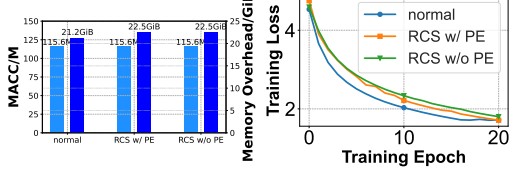

Original    Black: SL    Black: adv    Black: blur    Black: RCS    White: SL    White: RCS    e=10: SL    e=10: RCS

Figure 4: The visualization effect of input reconstruction on CelebA under black-box attacks (Black), white-box attacks (White) and attacks to training (e = 10).

The white-box attack follows Eq. 3 with 100,000 optimization iterations. As we can tell from Tab. 4, the transform-based methods successfully defends against the white-box attack for introducing randomness, so as our method, but their methods suffer great accuracy losses. This is also verified in the visualization results of Fig. 4, and further results on Cifar10 are left to Appendix C.5.

**Defence in training.** We launch an adaptive black-box attack on features collected over 10 rounds ($e = 10$ in Eq. 2). The attack we launch is much stronger than practice, as we choose $X_{aux}$ from the training set. In real-world, the adversary hardly obtains the training data. The final testing accuracy and privacy performance is reported in Tab. 5. It can be found that our RCS is strong in preserving the privacy of the training data without any loss of accuracy.

Table 5: Results of black-box attack on CelebA to the training process.

| | Utility | Privacy | | |
|---|---|---|---|---|
| | Accuracy↑ | SSIM↓ | PSNR↓ | F-SIM↓ |
| blur | 89.84% | 0.34 | 13.60 | 0.39 |
| Our RS | 91.40% | 0.27 | 10.81 | 0.39 |
| Our RCS | 89.92% | 0.09 | 10.46 | 0.34 |

Figure 5: The computational overhead, memory cost, and convergence curves on ImageNet of the normal split learning and our method. PE stands for position embedding (ViT-Base).

**Efficiency.** To see if our method incurs additional overhead to the normal split learning framework, we evaluate its efficiency by MAC operation counts (Macc, recording the number of multiplication and addition operations) on the edge, the memory consumption, and the convergence curve of pretraining ImageNet. The ImageNet image of size $224 \times 224$ is adopted with a batch size $256$. Results of Fig. 5 can be concluded that our methods are almost as efficient as the normal SL.

## 7 CONCLUSION

We propose a blind training method to realize privacy-preserving split learning, where the cloud trains over unknown data and model for the edge. The method is founded on the shuffle invariance property we discovered on Transformers, and other Transformer-based models. Theoretical proofs, property verification, and real-world performance resisting attacks are provided. Our method successfully defends black-box, and white-box attacks without degrading accuracy and efficiency.

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

---

**Algorithm 1** Shuffling at the Edge

---

1: Initialization: Initialize the model. Load permutation matrix $\boldsymbol{P}_C$ of size $(d, d)$ as the key and get its inverse $\boldsymbol{P}_C^{-1}$
2: Start training
3: **repeat**
4:     Start a new epoch
5:     **repeat**
6:         Get a batch of data $\boldsymbol{X}$ from data loader
7:         Get the patch embedding $\boldsymbol{Z}$ of size $(batch\_size, p, d)$.
8:         **if** using row shuffle **then**
9:             Get a random permutation matrix $\boldsymbol{P}_R$ of size $(p, p)$ and its inverse $\boldsymbol{P}_R^{-1}$
10:             $\boldsymbol{Z} = \text{torch.matmul}(\boldsymbol{P}_R, \boldsymbol{Z})$
11:         **end if**
12:         **if** using column shuffle **then**
13:             $\boldsymbol{Z} = \text{torch.matmul}(\boldsymbol{Z}, \boldsymbol{P}_C^{-1})$
14:         **end if**
15:         Send $\boldsymbol{Z}$ to the cloud and retrieve the output $\boldsymbol{Y}$
16:         **if** using row shuffle **then**
17:             $\boldsymbol{Y} = \text{torch.matmul}(\boldsymbol{P}_R^{-1}, \boldsymbol{Y})$
18:         **end if**
19:         **if** using column shuffle **then**
20:             $\boldsymbol{Y} = \text{torch.matmul}(\boldsymbol{Y}, \boldsymbol{P}_C)$
21:         **end if**
22:         Complete the usual backward propagation
23:     **until** done all batches
24: **until** done all epochs

---

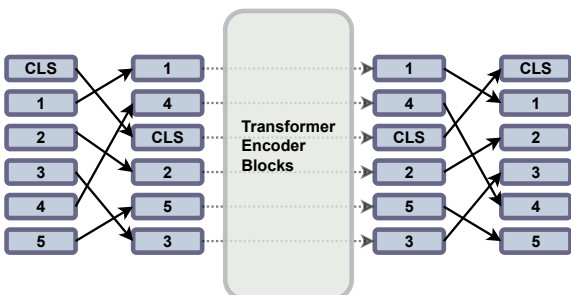

Figure 6: The row shuffle and unshuffle

## A  DETAILS ON SHUFFLING

Our shuffling scheme is described in pseudo code in Alg. 1 and the illustration figure is given by Fig. 6. It should be noted that the shuffling takes place not on the dimension of 'batches' but on the rest two dimensions. Taking ViT for example, each image is transformed into a $(p, d)$ matrix representing $p$ patches, and each patch denotes a fraction of the image. Each fraction is embedded into a $d$-dimensional vector. For example, let $\boldsymbol{Z}$ of shape $(3, 4)$ and the row shuffle matrix $\boldsymbol{P}_R$ be

$$\boldsymbol{Z} = \begin{pmatrix} 1 & 2 & 3 & 4 \\ 5 & 6 & 7 & 8 \\ 9 & 10 & 11 & 12 \end{pmatrix} \quad \boldsymbol{P}_R = \begin{pmatrix} 0 & 1 & 0 \\ 0 & 0 & 1 \\ 1 & 0 & 0 \end{pmatrix}.$$

The row shuffle is:

$$\boldsymbol{P}_R \boldsymbol{Z} = \begin{pmatrix} 5 & 6 & 7 & 8 \\ 9 & 10 & 11 & 12 \\ 1 & 2 & 3 & 4 \end{pmatrix}.$$

To further demonstrate the results of shuffling, we visualize the row shuffle and row-column shuffle methods on CelebA pictures in Fig. 7b and 7c. The column shuffle mixes up pixels from three channels and hence makes the images look like a gray one.

The shuffle method is simple, easy-to-deploy, and effective. We show why it works in the following section.

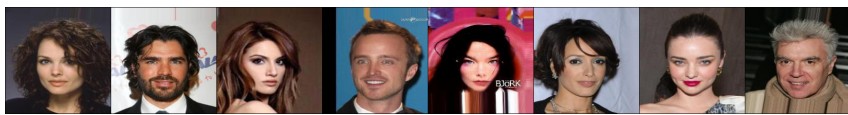

(a) Original CelebA pictures.

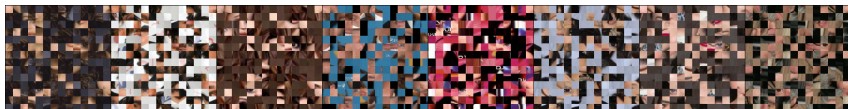

(b) Row (patch) shuffled pictures of CelebA. Each row of $\boldsymbol{Z}$ denotes a fraction of the image, and thus to shuffle the rows is to shuffle the fractions.

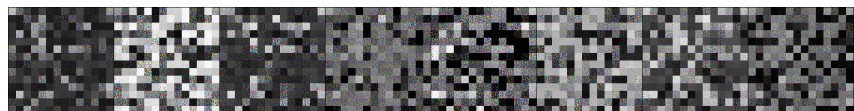

(c) Row-column shuffled pictures of CelebA. Each element in a patch comes from a pixel, and thus to shuffle the columns is to shuffle the pixels of three channels within the fraction.

Figure 7: Visualization of Shuffling. Our method works on embedded features instead of images, but here we directly shuffle the image to visualize shuffling.

# B PROOFS

We show the theorems in Sec. 4 hold for Transformer encoder blocks.

## B.1 NOTATIONS AND LEMMAS

The Transformer encoder block is denoted as $Enc$ and the loss is $\ell$. The patch embedding of a single input $\boldsymbol{X}$ is expressed as $\boldsymbol{Z}$ of shape $(p, d)$. The first layer in the self-attention contains three parallel linear layers projecting $\boldsymbol{Z}$ to $Q, K, V$ as

$$\boldsymbol{Z}\boldsymbol{W}_Q^T = \boldsymbol{Q}, \tag{13}$$

$$\boldsymbol{Z}\boldsymbol{W}_K^T = \boldsymbol{K}, \tag{14}$$

$$\boldsymbol{Z}\boldsymbol{W}_V^T = \boldsymbol{V}. \tag{15}$$

$Q, K, V$ are fed to the following attention operation

$$\boldsymbol{S} = Softmax(\frac{\boldsymbol{Q}\boldsymbol{K}^T}{\sqrt{d}}), \tag{16}$$

$$\boldsymbol{A} = \boldsymbol{S}\boldsymbol{V}, \tag{17}$$

where $\boldsymbol{S}$ and $\boldsymbol{A}$ are the softmax output, and the attention output, respectively.

The part following the attention layer is the MLP layer:

$$\boldsymbol{A}_1 = \boldsymbol{A}\boldsymbol{W}_1^T, \tag{18}$$

$$\boldsymbol{H} = a(\boldsymbol{A}_1), \tag{19}$$

$$\boldsymbol{A}_2 = \boldsymbol{H}\boldsymbol{W}_2^T \tag{20}$$

where $\boldsymbol{A}_1$, $\boldsymbol{A}_2$ are the outputs of the linear layers with weights $\boldsymbol{W}_1, \boldsymbol{W}_2$, respectively, and $\boldsymbol{H}$ is the output of the element-wise activation function $a$, being ReLu or Tanh.

**Element-wise operators** including shortcut, Hadamard product, matrix addition/subtraction and other element-wise functions are permutation-equivalent:

**Lemma 1.** *Element-wise operators are permutation-equivalent:*

$$(\boldsymbol{P}_1 \boldsymbol{A} \boldsymbol{P}_2) \odot (\boldsymbol{P}_1 \boldsymbol{B} \boldsymbol{P}_2) = \boldsymbol{P}_1 (\boldsymbol{A} \odot \boldsymbol{B}) \boldsymbol{P}_2. \tag{21}$$

On the left hand-side of the equation, $a_{ij}$ in $\boldsymbol{A}$ and $b_{ij}$ in $\boldsymbol{B}$ are permuted to the same position before being performed the operation. On the right hand-side, $a_{ij}$ and $b_{ij}$ are performed the operation of which the results are permuted. The two are obviously equivalent. Importantly,

**Lemma 2.** *Softmax is permutation-equivalent:*

$$Softmax(\boldsymbol{P}_1 \boldsymbol{A} \boldsymbol{P}_2) = \boldsymbol{P}_1 Softmax(\boldsymbol{A}) \boldsymbol{P}_2. \tag{22}$$

This is because an element is always normalized with the same group of elements, which are not changed in permutations. Thus Softmax is permutation-equivalent.

### B.2 PROOF OF THEOREM 1

*Proof.* We prove the row-permutation equivalence of Transformer encoder blocks in forward propagation (Eq. 5). We denote the row-permutation matrix as $\boldsymbol{P}_R$. Features are permuted as $\boldsymbol{Z}_{(R)} = \boldsymbol{P}_R \boldsymbol{Z}$. It is worth noting that

$$\boldsymbol{P}_R^T \boldsymbol{P}_R = \boldsymbol{E}$$

holds for permutation matrix $\boldsymbol{P}_R$. $\boldsymbol{E}$ is the identity matrix.

We first prove the permutation equivalence of attention.

$$\begin{aligned}
\boldsymbol{A}_{(R)} &= Softmax(\frac{\boldsymbol{Q}_{(R)} \boldsymbol{K}_{(R)}^T}{\sqrt{d}}) \boldsymbol{V}_{(R)} \\
&= Softmax(\frac{\boldsymbol{Z}_{(R)} \boldsymbol{W}_Q^T \boldsymbol{W}_K \boldsymbol{Z}_{(R)}^T}{\sqrt{d}}) \boldsymbol{Z}_{(R)} \boldsymbol{W}_V^T \\
&= Softmax(\frac{\boldsymbol{P}_R \boldsymbol{Z} \boldsymbol{W}_Q^T \boldsymbol{W}_K \boldsymbol{Z}^T \boldsymbol{P}_R^T}{\sqrt{d}}) \boldsymbol{P}_R \boldsymbol{Z} \boldsymbol{W}_V^T \\
&= \boldsymbol{P}_R Softmax(\frac{\boldsymbol{Z} \boldsymbol{W}_Q^T \boldsymbol{W}_K \boldsymbol{Z}^T}{\sqrt{d}}) \boldsymbol{P}_R^T \boldsymbol{P}_R \boldsymbol{Z} \boldsymbol{W}_V^T \\
&= \boldsymbol{P}_R Softmax(\frac{\boldsymbol{Z} \boldsymbol{W}_Q^T \boldsymbol{W}_K \boldsymbol{Z}^T}{\sqrt{d}}) \boldsymbol{Z} \boldsymbol{W}_V^T \\
&= \boldsymbol{P}_R Softmax(\frac{\boldsymbol{Q} \boldsymbol{K}^T}{\sqrt{d}}) \boldsymbol{V} \\
&= \boldsymbol{P}_R \boldsymbol{A},
\end{aligned}$$

and thus

$$Attention(\boldsymbol{P}_R \boldsymbol{Z}) = \boldsymbol{P}_R Attention(\boldsymbol{Z}). \tag{23}$$

By Lemma 2, the softmax layer following the attention is permutation-equivalent. And the subsequent is the MLP layer which satisfies:

$$\begin{aligned}
\boldsymbol{A}_{2(R)} &= a(\boldsymbol{A}_{(R)} \boldsymbol{W}_1^T) \boldsymbol{W}_2^T \\
&= a(\boldsymbol{P}_R \boldsymbol{A} \boldsymbol{W}_1^T) \boldsymbol{W}_2^T \\
&= \boldsymbol{P}_R a(\boldsymbol{A} \boldsymbol{W}_1^T) \boldsymbol{W}_2^T \\
&= \boldsymbol{P}_R \boldsymbol{A}_2,
\end{aligned}$$

meaning

$$MLP(\boldsymbol{P}_R \boldsymbol{A}) = \boldsymbol{P}_R MLP(\boldsymbol{A}). \tag{24}$$

Hence we have proved the transformer encoder block is row-permutation-equivalent:

$$Enc(\boldsymbol{P}_R \boldsymbol{Z}) = \boldsymbol{P}_R Enc(\boldsymbol{Z}). \tag{25}$$

$\square$

### B.3 PROOF OF THEOREM 2

*Proof.* To prove the gradients are the same when the inputs are row-shuffled, we first calculate the all the gradients from the final layer back to the first. Gradients are expressed as

$$
\begin{aligned}
\mathrm{d}l &= \mathrm{tr}(\frac{\partial l}{\partial \boldsymbol{A}_2}^T \mathrm{d}\boldsymbol{A}_2) \\
&= \mathrm{tr}(\frac{\partial l}{\partial \boldsymbol{A}_2}^T (\mathrm{d}\boldsymbol{H})\boldsymbol{W}_2^T) + \mathrm{tr}(\frac{\partial l}{\partial \boldsymbol{A}_2}^T \boldsymbol{H}\mathrm{d}(\boldsymbol{W}_2^T)).
\end{aligned}
$$

Let's study $\boldsymbol{H}$ first:

$$
\begin{aligned}
\mathrm{d}l_1 &\triangleq \mathrm{tr}(\frac{\partial l}{\partial \boldsymbol{A}_2}^T (\mathrm{d}\boldsymbol{H})\boldsymbol{W}_2^T) \\
&= \mathrm{tr}(\boldsymbol{W}_2^T \frac{\partial l}{\partial \boldsymbol{A}_2}^T \mathrm{d}\boldsymbol{H}) \\
&= \mathrm{tr}((\frac{\partial l}{\partial \boldsymbol{A}_2}\boldsymbol{W}_2)^T \mathrm{d}\boldsymbol{H}),
\end{aligned}
$$

indicating

$$
\frac{\partial l}{\partial \boldsymbol{H}} = \frac{\partial l}{\partial \boldsymbol{A}_2}\boldsymbol{W}_2. \tag{26}
$$

For $\boldsymbol{W}_2$,

$$
\begin{aligned}
\mathrm{d}l_2 &\triangleq \mathrm{tr}(\frac{\partial l}{\partial \boldsymbol{A}_2}^T \boldsymbol{H}\mathrm{d}(\boldsymbol{W}_2^T)) \\
&= \mathrm{tr}(\mathrm{d}\boldsymbol{W}_2 \boldsymbol{H}^T \frac{\partial l}{\partial \boldsymbol{A}_2}) \\
&= \mathrm{tr}((\frac{\partial l}{\partial \boldsymbol{A}_2}^T \boldsymbol{H})^T \mathrm{d}\boldsymbol{W}_2),
\end{aligned}
$$

and

$$
\frac{\partial l}{\partial \boldsymbol{W}_2} = \frac{\partial l}{\partial \boldsymbol{A}_2}^T \boldsymbol{H}. \tag{27}
$$

For $\boldsymbol{A}_1$:

$$
\begin{aligned}
\mathrm{d}l_1 &= \mathrm{tr}((\frac{\partial l}{\partial \boldsymbol{A}_2}\boldsymbol{W}_2)^T \mathrm{d}\boldsymbol{H}) \\
&= \mathrm{tr}(\frac{\partial l}{\partial \boldsymbol{H}}^T \mathrm{d}(a(\boldsymbol{A}_1))) \\
&= \mathrm{tr}(\frac{\partial l}{\partial \boldsymbol{H}}^T a'(\boldsymbol{A}_1) \odot \mathrm{d}\boldsymbol{A}_1)) \\
&= \mathrm{tr}((\frac{\partial l}{\partial \boldsymbol{H}} \odot a'(\boldsymbol{A}_1))^T \mathrm{d}\boldsymbol{A}_1),
\end{aligned}
$$

and

$$
\frac{\partial l}{\partial \boldsymbol{A}_1} = \frac{\partial l}{\partial \boldsymbol{A}_2}\boldsymbol{W}_2 \odot a'(\boldsymbol{A}_1). \tag{28}
$$

Similarly, we calculate the gradients of $\boldsymbol{A}$ and $\boldsymbol{W}_1$:

$$
\frac{\partial l}{\partial \boldsymbol{A}} = \frac{\partial l}{\partial \boldsymbol{A}_1}\boldsymbol{W}_1, \tag{29}
$$

$$
\frac{\partial l}{\partial \boldsymbol{W}_1} = \frac{\partial l}{\partial \boldsymbol{A}_1}^T \boldsymbol{A}. \tag{30}
$$

In the attention operation:

$$
\begin{aligned}
\mathrm{d}l_3 &\triangleq \mathrm{tr}(\frac{\partial l}{\partial \boldsymbol{A}}^T \mathrm{d}\boldsymbol{A}) \\
&= \mathrm{tr}(\frac{\partial l}{\partial \boldsymbol{A}}^T (\mathrm{d}\boldsymbol{S})\boldsymbol{V}) + \mathrm{tr}(\frac{\partial l}{\partial A}^T \boldsymbol{S}\mathrm{d}\boldsymbol{V}) \\
&= \mathrm{tr}((\frac{\partial l}{\partial \boldsymbol{A}}\boldsymbol{V}^T)^T\mathrm{d}\boldsymbol{S}) + \mathrm{tr}((\boldsymbol{S}^T\frac{\partial l}{\partial \boldsymbol{A}})^T\mathrm{d}\boldsymbol{V}),
\end{aligned}
$$

and

$$
\frac{\partial l}{\partial \boldsymbol{S}} = \frac{\partial l}{\partial \boldsymbol{A}}\boldsymbol{V}^T, \tag{31}
$$

$$
\frac{\partial l}{\partial \boldsymbol{V}} = \boldsymbol{S}^T\frac{\partial l}{\partial \boldsymbol{A}}. \tag{32}
$$

First, for $\boldsymbol{V} = \boldsymbol{Z}\boldsymbol{W}_V^T$:

$$
\begin{aligned}
\mathrm{d}l_4 &\triangleq \mathrm{tr}(\frac{\partial l}{\partial \boldsymbol{V}}^T \mathrm{d}\boldsymbol{V}) \\
&= \mathrm{tr}(\frac{\partial l}{\partial \boldsymbol{V}}^T (\mathrm{d}\boldsymbol{Z})\boldsymbol{W}_V^T) + \mathrm{tr}(\frac{\partial l}{\partial \boldsymbol{V}}^T \boldsymbol{Z}\mathrm{d}\boldsymbol{W}_V^T).
\end{aligned}
$$

Similarly, the gradients of $\boldsymbol{Z}$ and $\boldsymbol{W}_V$ are:

$$
\frac{\partial l}{\partial \boldsymbol{Z}} = \frac{\partial l}{\partial \boldsymbol{V}}\boldsymbol{W}_V, \tag{33}
$$

$$
\frac{\partial l}{\partial \boldsymbol{W}_V} = \frac{\partial l}{\partial \boldsymbol{V}}^T \boldsymbol{Z}. \tag{34}
$$

Now we focus on $\boldsymbol{S} = Softmax(\frac{\boldsymbol{Q}\boldsymbol{K}^T}{\sqrt{d}})$:

$$
\begin{aligned}
\mathrm{d}l_5 &\triangleq \mathrm{tr}(\frac{\partial l}{\partial \boldsymbol{S}}^T \mathrm{d}\boldsymbol{S}) \\
&= \mathrm{tr}(\frac{\partial l}{\partial \boldsymbol{S}}^T (diag(\boldsymbol{S}) - \boldsymbol{S}^T\boldsymbol{S})\mathrm{d}(\frac{\boldsymbol{Q}\boldsymbol{K}^T}{\sqrt{d}})) \\
&= \mathrm{tr}(((diag(\boldsymbol{S}) - \boldsymbol{S}^T\boldsymbol{S})^T\frac{\partial l}{\partial \boldsymbol{S}})^T\mathrm{d}(\frac{\boldsymbol{Q}\boldsymbol{K}^T}{\sqrt{d}})),
\end{aligned}
$$

and thus

$$
\frac{\partial l}{\partial \boldsymbol{Q}} = \frac{1}{\sqrt{d}}((diag(\boldsymbol{S}) - \boldsymbol{S}^T\boldsymbol{S})^T\frac{\partial l}{\partial \boldsymbol{S}})\boldsymbol{K}, \tag{35}
$$

$$
\frac{\partial l}{\partial \boldsymbol{K}} = \frac{1}{\sqrt{d}}((diag(\boldsymbol{S}) - \boldsymbol{S}^T\boldsymbol{S})^T\frac{\partial l}{\partial \boldsymbol{S}})^T\boldsymbol{Q}. \tag{36}
$$

And similarly the gradients of $\boldsymbol{W}_Q$ and $\boldsymbol{W}_K$ are:

$$
\frac{\partial l}{\partial \boldsymbol{W}_Q} = \frac{\partial l}{\partial \boldsymbol{Q}}^T \boldsymbol{Z}, \tag{37}
$$

$$
\frac{\partial l}{\partial \boldsymbol{W}_K} = \frac{\partial l}{\partial \boldsymbol{K}}^T \boldsymbol{Z}. \tag{38}
$$

**What happens in backward propagation with RS:** First, let us take a black-box view of the Transformer encoder blocks. In our scheme, the input of $Enc$ is the shuffled smashed data $\boldsymbol{P}_R\boldsymbol{Z}$ and the output is $\boldsymbol{P}_R\boldsymbol{A}_2$. The edge client obtains $\boldsymbol{P}_R\boldsymbol{A}_2$ and reverse the shuffling order to get

$\boldsymbol{A}_{3(R)} = \boldsymbol{P}_R^T \boldsymbol{A}_{2(R)}$. In the following, we denote all variables involved with our RS method with subscript $(R)$, and variables in vanilla split learning are without the subscript. According to Eq. 1,

$$\boldsymbol{A}_{3(R)} = \boldsymbol{P}_R^T \boldsymbol{A}_{2(R)} = \boldsymbol{P}_R^T \boldsymbol{P}_R \boldsymbol{A}_2 = \boldsymbol{A}_2. \tag{39}$$

We have $\boldsymbol{A}_3 = \boldsymbol{A}_2$ in plain SL, and thus

$$\frac{\partial l}{\partial \boldsymbol{A}_{3(R)}} = \frac{\partial l}{\partial \boldsymbol{A}_2} = \frac{\partial l}{\partial \boldsymbol{A}_3}. \tag{40}$$

Eq. 40 suggests since the forwarding (from $\boldsymbol{A}_{3(R)}$ to the loss) is the same between the original and our scheme on the edge, its backward process is also equivalent, *i.e.,* each gradient of the weights between $\boldsymbol{A}_{3(R)}$ and the loss $l$ is the same for our method and the original SL. Hence we only need to focus on gradients from $\boldsymbol{A}_{2(R)}$ backward.

$$\mathrm{d}l = \mathrm{tr}(\frac{\partial l}{\partial \boldsymbol{A}_{3(R)}}^T \boldsymbol{P}_R^T \mathrm{d}\boldsymbol{A}_{2(R)})$$

$$= \mathrm{tr}((\boldsymbol{P}_R \frac{\partial l}{\partial \boldsymbol{A}_3})^T \mathrm{d}\boldsymbol{A}_{2(R)}),$$

and thus

$$\frac{\partial l}{\partial \boldsymbol{A}_{2(R)}} = \boldsymbol{P}_R \frac{\partial l}{\partial \boldsymbol{A}_3} = \boldsymbol{P}_R \frac{\partial l}{\partial \boldsymbol{A}_2}. \tag{41}$$

It is quite an interesting conclusion, and we will soon find it important to the permutation-invariance of gradients of the weight matrix. By substituting Eq. 41 into Eq. 27, we obtain:

$$\frac{\partial l}{\partial \boldsymbol{W}_{2(R)}} = \frac{\partial l}{\partial \boldsymbol{A}_{2(R)}}^T \boldsymbol{H}_{(R)} \tag{42}$$

$$= \frac{\partial l}{\partial \boldsymbol{A}_2}^T \boldsymbol{P}_R^T \boldsymbol{P}_R \boldsymbol{H} \tag{43}$$

$$= \frac{\partial l}{\partial \boldsymbol{A}_2}^T \boldsymbol{H} \tag{44}$$

$$= \frac{\partial l}{\partial \boldsymbol{W}_2}. \tag{45}$$

Eq. 45 reveals that the gradients of the weight matrix in our RS scheme would be the same as the gradients in vanilla SL. In fact, each gradient of the weight matrix is 'permutation-invariant,' i.e., satisfying Eq. 45, whereas each gradient of the intermediate feature is 'permutation-equivalent,' meeting Eq. 41. Properties of Eq. 45 and Eq. 41 are transductive, carrying from the final layer backward to the first layer. Hence by Eq. 26,

$$\frac{\partial l}{\partial \boldsymbol{H}_{(R)}} = \boldsymbol{P}_R \frac{\partial l}{\partial \boldsymbol{H}}, \tag{46}$$

and by Eq. 28,

$$\frac{\partial l}{\partial \boldsymbol{A}_{1(R)}} = \boldsymbol{P}_R \frac{\partial l}{\partial \boldsymbol{A}_2} \boldsymbol{W}_2 \odot \boldsymbol{P}_R a'(\boldsymbol{A}_1)$$

$$= \boldsymbol{P}_R(\frac{\partial l}{\partial \boldsymbol{A}_2} \boldsymbol{W}_2 \odot a'(\boldsymbol{A}_1)).$$

Thus,

$$\frac{\partial l}{\partial \boldsymbol{A}_{1(R)}} = \boldsymbol{P}_R \frac{\partial l}{\partial \boldsymbol{A}_1}, \tag{47}$$

and by Eq. 30,

$$\frac{\partial l}{\partial \boldsymbol{W}_{1(R)}} = \frac{\partial l}{\partial \boldsymbol{W}_1}. \tag{48}$$

By Eq. 29, we have:

$$\frac{\partial l}{\partial \boldsymbol{A}_{(R)}} = \boldsymbol{P}_R \frac{\partial l}{\partial \boldsymbol{A}} \tag{49}$$

which passes the property to $\boldsymbol{S}$. But $\boldsymbol{S}$ is a tricky one since in the derivation of Eq. 23, we notice that $\boldsymbol{S}_{(R)} = \boldsymbol{P}_R \boldsymbol{S} \boldsymbol{P}_R^T$. According to Eq. 31,

$$\frac{\partial l}{\partial \boldsymbol{S}_{(R)}} = \frac{\partial l}{\partial \boldsymbol{A}_{(R)}} \boldsymbol{V}_{(R)}^T$$

$$= \boldsymbol{P}_R \frac{\partial l}{\partial \boldsymbol{A}} \boldsymbol{V}^T \boldsymbol{P}_R^T,$$

which interestingly leads to

$$\frac{\partial l}{\partial \boldsymbol{S}_{(R)}} = \boldsymbol{P}_R \frac{\partial l}{\partial \boldsymbol{S}} \boldsymbol{P}_R^T. \tag{50}$$

This is consistent with the $\boldsymbol{S}_{(R)}$'s permutation form, which passes the elegant properties down to $\boldsymbol{Q}, \boldsymbol{K}$:

$$\frac{\partial l}{\partial \boldsymbol{Q}_{(R)}} = \frac{1}{\sqrt{d}}((diag(\boldsymbol{S}_{(R)}) - \boldsymbol{S}_{(R)}^T \boldsymbol{S}_{(R)})^T \frac{\partial l}{\partial \boldsymbol{S}_{(R)}})\boldsymbol{K}_{(R)}$$

$$= \frac{1}{\sqrt{d}}((\boldsymbol{P}_R diag(\boldsymbol{S})\boldsymbol{P}_R^T - \boldsymbol{P}_R \boldsymbol{S}^T \boldsymbol{P}_R^T \boldsymbol{P}_R \boldsymbol{S} \boldsymbol{P}_R^T)^T \boldsymbol{P}_R \frac{\partial l}{\partial \boldsymbol{S}} \boldsymbol{P}_R^T)\boldsymbol{P}_R \boldsymbol{K}$$

$$= \frac{1}{\sqrt{d}}(\boldsymbol{P}_R(diag(\boldsymbol{S}) - \boldsymbol{S}^T \boldsymbol{S})^T \boldsymbol{P}_R^T \boldsymbol{P}_R \frac{\partial l}{\partial \boldsymbol{S}} \boldsymbol{P}_R^T)\boldsymbol{P}_R \boldsymbol{K}$$

$$= \boldsymbol{P}_R \frac{1}{\sqrt{d}}((diag(\boldsymbol{S}) - S^T S)^T \frac{\partial l}{\partial \boldsymbol{S}})\boldsymbol{K},$$

and hence

$$\frac{\partial l}{\partial \boldsymbol{Q}_{(R)}} = \boldsymbol{P}_R \frac{\partial l}{\partial \boldsymbol{Q}}. \tag{51}$$

Similarly,

$$\frac{\partial l}{\partial \boldsymbol{K}_{(R)}} = \boldsymbol{P}_R \frac{\partial l}{\partial \boldsymbol{K}}. \tag{52}$$

By Eq. 37,

$$\frac{\partial l}{\partial \boldsymbol{W}_{Q(R)}} = \frac{\partial l}{\partial \boldsymbol{Q}_{(R)}}^T \boldsymbol{Z}_{(R)} \tag{53}$$

$$= \frac{\partial l}{\partial \boldsymbol{Q}}^T \boldsymbol{P}_R^T \boldsymbol{P}_R \boldsymbol{Z} = \frac{\partial l}{\partial \boldsymbol{W}_Q}. \tag{54}$$

Similarly,

$$\frac{\partial l}{\partial \boldsymbol{W}_{K(R)}} = \frac{\partial l}{\partial \boldsymbol{W}_K}. \tag{55}$$

And for $\boldsymbol{V}$,

$$\frac{\partial l}{\partial \boldsymbol{V}_{(R)}} = \boldsymbol{S}_{(R)}^T \frac{\partial l}{\partial \boldsymbol{A}_{(R)}} \tag{56}$$

$$= \boldsymbol{P}_R \boldsymbol{S}^T \boldsymbol{P}_R^T \cdot \boldsymbol{P}_R \frac{\partial l}{\partial \boldsymbol{A}} \tag{57}$$

$$= \boldsymbol{P}_R \boldsymbol{S}^T \frac{\partial l}{\partial \boldsymbol{A}} \tag{58}$$

$$= \boldsymbol{P}_R \frac{\partial l}{\partial \boldsymbol{V}} \tag{59}$$

Similarly,

$$\frac{\partial l}{\partial \boldsymbol{W}_{V(R)}} = \frac{\partial l}{\partial \boldsymbol{V}_{(R)}}^T \boldsymbol{X}_{(R)} \tag{60}$$

$$= \frac{\partial l}{\partial \boldsymbol{V}} \boldsymbol{P}_R^T \cdot \boldsymbol{P}_R \boldsymbol{X} \tag{61}$$

$$= \frac{\partial l}{\partial \boldsymbol{W}_V}. \tag{62}$$

For now, we have proved that the gradients of weights in our RS scheme are exactly the same as the gradients of weights in vanilla split learning, and by induction, we can conclude that the learned $Enc$ with RS method is no different from the learned $Enc$ without. It is worth mentioning that if the attention is cut to multi-head, the property remains because cutting the second dimension (column) does not affect the permutation of the first dimension (row). □

### B.4 Proof of Theorem 3

*Proof.* We denote variables involved in our RCS method with subscript $(P)$. Row shuffling is included as a special case of RCS.

First and foremost, we 'encrypt' all the weight matrices by Eq. 10:

$$\boldsymbol{W}_{i(p)} = \boldsymbol{P}_C \boldsymbol{W}_i \boldsymbol{P}_C^T,$$

where $\boldsymbol{P}_C$ is the column permutation matrix, $\boldsymbol{W}_i$ is the weight of a normal $Enc$, and $i \in \{1, 2, Q, K, V\}$. We denote the Transformer encoder block with such 'encryption' as $Enc_{(P)}$. Note that this operation requires $\boldsymbol{W}_i$ to be a square one. We have proposed two ways to achieve this with little modification to the original model without performance loss.

For $\boldsymbol{Q}$:

$$\boldsymbol{Q}_{(P)} = \boldsymbol{Z}_{(P)} \boldsymbol{W}_{\boldsymbol{Q}(P)}^T \tag{63}$$

$$= \boldsymbol{P}_R \boldsymbol{Z} \boldsymbol{P}_C^T \cdot \boldsymbol{P}_C \boldsymbol{W}_Q^T \boldsymbol{P}_C^T \tag{64}$$

$$= \boldsymbol{P}_R \boldsymbol{Z} \boldsymbol{W}_Q^T \boldsymbol{P}_C^T \tag{65}$$

$$= \boldsymbol{P}_R \boldsymbol{Q} \boldsymbol{P}_C^T. \tag{66}$$

Similarly for $\boldsymbol{K}, \boldsymbol{V}$:

$$\boldsymbol{K}_{(P)} = \boldsymbol{P}_R \boldsymbol{K} \boldsymbol{P}_C^T, \tag{67}$$

$$\boldsymbol{V}_{(P)} = \boldsymbol{P}_R \boldsymbol{V} \boldsymbol{P}_C^T. \tag{68}$$

For $\boldsymbol{S} = Softmax(\frac{\boldsymbol{Q}\boldsymbol{K}^T}{\sqrt{d}})$:

$$\boldsymbol{S}_{(P)} = Softmax(\frac{\boldsymbol{Q}_{(P)} \boldsymbol{K}_{(P)}^T}{\sqrt{d}}) \tag{69}$$

$$= Softmax(\frac{\boldsymbol{P}_R \boldsymbol{Q} \boldsymbol{P}_C^T \cdot \boldsymbol{P}_C \boldsymbol{K}^T \boldsymbol{P}_R^T}{\sqrt{d}}) \tag{70}$$

$$= Softmax(\frac{\boldsymbol{P}_R \boldsymbol{Q} \boldsymbol{K}^T \boldsymbol{P}_R^T}{\sqrt{d}}) \tag{71}$$

$$= \boldsymbol{P}_R Softmax(\frac{\boldsymbol{Q}\boldsymbol{K}^T}{\sqrt{d}}) \boldsymbol{P}_R^T \tag{72}$$

$$= \boldsymbol{P}_R \boldsymbol{S} \boldsymbol{P}_R^T. \tag{73}$$

So for $\boldsymbol{A}$:

$$\boldsymbol{A}_{(P)} = \boldsymbol{S}_{(P)} \boldsymbol{V}_{(P)} \tag{74}$$

$$= \boldsymbol{P}_R \boldsymbol{S} \boldsymbol{P}_R^T \cdot \boldsymbol{P}_R \boldsymbol{V} \boldsymbol{P}_C^T \tag{75}$$

$$= \boldsymbol{P}_R \boldsymbol{S} \boldsymbol{V} \boldsymbol{P}_C^T \tag{76}$$

$$= \boldsymbol{P}_R \boldsymbol{A} \boldsymbol{P}_C^T. \tag{77}$$

Following the attention layer, $\boldsymbol{A}$ is fed to the MLP layer:

$$\boldsymbol{A}_{1(P)} = \boldsymbol{A}_{(P)}\boldsymbol{W}_{1(P)}^T \tag{78}$$

$$= \boldsymbol{P}_R\boldsymbol{A}\boldsymbol{P}_C^T \cdot \boldsymbol{P}_C\boldsymbol{W}_1\boldsymbol{P}_C^T \tag{79}$$

$$= \boldsymbol{P}_R\boldsymbol{A}\boldsymbol{W}_1\boldsymbol{P}_C^T \tag{80}$$

$$= \boldsymbol{P}_R\boldsymbol{A}_1\boldsymbol{P}_C^T. \tag{81}$$

Similarly for $\boldsymbol{A}_2$,

$$\boldsymbol{A}_{2(P)} = \boldsymbol{P}_R\boldsymbol{A}_2\boldsymbol{P}_C^T. \tag{82}$$

As for the activation in the middle, the element-wise activation function is permutation-equivalent:

$$\boldsymbol{H}_{(P)} = \boldsymbol{P}_R\boldsymbol{H}\boldsymbol{P}_C^T. \tag{83}$$

Overall, we have proved Thm. 3. $\qquad\square$

## B.5 PROOF OF THEOREM 4

*Proof.* By row and column unshuffle, the computation of the forward propagation and backward propagation on the edge is no different with or without our RCS method. Hence we only focus on the propagation of the Transformer encoder blocks.

Similarly to the proof on RS, we denote $\boldsymbol{A}_{3(P)}$ as the reversed intermediate feature that the edge receives:

$$\boldsymbol{A}_{3(P)} = \boldsymbol{P}_R^T\boldsymbol{A}_{2(P)}\boldsymbol{P}_C. \tag{84}$$

By Thm. 3, we have

$$\boldsymbol{A}_{3(P)} = \boldsymbol{A}_2 = \boldsymbol{A}_3. \tag{85}$$

First we focuses on the MLP layer:

$$\mathrm{d}l = \mathrm{tr}(\frac{\partial l}{\partial \boldsymbol{A}_3}^T \boldsymbol{P}_R^T \mathrm{d}(\boldsymbol{A}_{2(P)})\boldsymbol{P}_C)$$

$$= \mathrm{tr}(\boldsymbol{P}_C\frac{\partial l}{\partial \boldsymbol{A}_3}^T \boldsymbol{P}_R^T \mathrm{d}\boldsymbol{A}_{2(P)})$$

$$= \mathrm{tr}((\boldsymbol{P}_R\frac{\partial l}{\partial \boldsymbol{A}_3}\boldsymbol{P}_C^T)^T \mathrm{d}\boldsymbol{A}_{2(P)}),$$

that is:

$$\frac{\partial l}{\partial \boldsymbol{A}_{2(P)}} = \boldsymbol{P}_R\frac{\partial l}{\partial \boldsymbol{A}_2}\boldsymbol{P}_C^T. \tag{86}$$

With $\boldsymbol{H}_{(P)} = \boldsymbol{P}_R\boldsymbol{H}\boldsymbol{P}_C^T$ and Eq. 27, the gradient:

$$\frac{\partial l}{\partial \boldsymbol{W}_{2(P)}} = \frac{\partial l}{\partial \boldsymbol{A}_{2(P)}}^T \boldsymbol{H}_{(P)}$$

$$= \boldsymbol{P}_C\frac{\partial l}{\partial \boldsymbol{A}_2}\boldsymbol{P}_R^T \cdot \boldsymbol{P}_R\boldsymbol{H}\boldsymbol{P}_C^T$$

$$= \boldsymbol{P}_C\frac{\partial l}{\partial \boldsymbol{A}_2}\boldsymbol{H}\boldsymbol{P}_C^T$$

$$= \boldsymbol{P}_C\frac{\partial l}{\partial \boldsymbol{W}_2}\boldsymbol{P}_C^T,$$

that is:

$$\frac{\partial l}{\partial \boldsymbol{W}_{2(P)}} = \boldsymbol{P}_C\frac{\partial l}{\partial \boldsymbol{W}_2}\boldsymbol{P}_C^T. \tag{87}$$

By Eq. 28, Eq. 87, and a similar derivation of Eq. 47, we have

$$\frac{\partial l}{\partial \boldsymbol{A}_{1(P)}} = \frac{\partial l}{\partial \boldsymbol{A}_{2(P)}} \boldsymbol{W}_{2(P)} \odot a'(\boldsymbol{A}_{1(P)})$$

$$= [\boldsymbol{P}_R \frac{\partial l}{\partial \boldsymbol{A}_2} \boldsymbol{P}_C^T \cdot \boldsymbol{P}_C \boldsymbol{W}_2 \boldsymbol{P}_C^T] \odot [\boldsymbol{P}_R a'(\boldsymbol{A}_1) \boldsymbol{P}_C^T]$$

$$= [\boldsymbol{P}_R \frac{\partial l}{\partial \boldsymbol{A}_2} \boldsymbol{W}_2 \boldsymbol{P}_C^T] \odot [\boldsymbol{P}_R a'(\boldsymbol{A}_1) \boldsymbol{P}_C^T]$$

$$= \boldsymbol{P}_R [\frac{\partial l}{\partial \boldsymbol{A}_2} \boldsymbol{W}_2 \odot a'(\boldsymbol{A}_1)] \boldsymbol{P}_C^T$$

$$= \boldsymbol{P}_R \frac{\partial l}{\partial \boldsymbol{A}_1} \boldsymbol{P}_C^T,$$

that is:

$$\frac{\partial l}{\partial \boldsymbol{A}_{1(P)}} = \boldsymbol{P}_R \frac{\partial l}{\partial \boldsymbol{A}_1} \boldsymbol{P}_C^T. \tag{88}$$

The weight $\boldsymbol{W}_{1(P)}$ in the MLP has the following gradient by Eq. 30:

$$\frac{\partial l}{\partial \boldsymbol{W}_{1(P)}} = \frac{\partial l}{\partial \boldsymbol{A}_{1(P)}}^T \boldsymbol{A}_{(P)}$$

$$= \boldsymbol{P}_C \frac{\partial l}{\partial \boldsymbol{A}_1}^T \boldsymbol{P}_R^T \cdot \boldsymbol{P}_R \boldsymbol{A} \boldsymbol{P}_C^T$$

$$= \boldsymbol{P}_C \frac{\partial l}{\partial \boldsymbol{W}_1} \boldsymbol{P}_C^T,$$

that is:

$$\frac{\partial l}{\partial \boldsymbol{W}_{1(P)}} = \boldsymbol{P}_C \frac{\partial l}{\partial \boldsymbol{W}_1} \boldsymbol{P}_C^T. \tag{89}$$

And we come to the attention operation, from Eq. 29, we have

$$\frac{\partial l}{\partial \boldsymbol{A}_{(P)}} = \frac{\partial l}{\partial \boldsymbol{A}_{1(P)}} \boldsymbol{W}_{1(P)}$$

$$= \boldsymbol{P}_R \frac{\partial l}{\partial \boldsymbol{A}_1} \boldsymbol{P}_C^T \cdot \boldsymbol{P}_C \boldsymbol{W}_1 \boldsymbol{P}_C^T$$

$$= \boldsymbol{P}_R \frac{\partial l}{\partial \boldsymbol{A}_1} \boldsymbol{W}_1 \boldsymbol{P}_C^T$$

$$= \boldsymbol{P}_R \frac{\partial l}{\partial \boldsymbol{A}} \boldsymbol{P}_C^T,$$

that is:

$$\frac{\partial l}{\partial \boldsymbol{A}_{(P)}} = \boldsymbol{P}_R \frac{\partial l}{\partial \boldsymbol{A}} \boldsymbol{P}_C^T. \tag{90}$$

Hence we observe the permutations rules for the gradients of the intermediate-layer outputs vary from the gradients of the weights. As for the gradients of the softmax-layer output, we have

$$\frac{\partial l}{\partial \boldsymbol{S}_{(P)}} = \frac{\partial l}{\partial \boldsymbol{A}_{(P)}} \boldsymbol{V}_{(P)}^T$$

$$= \boldsymbol{P}_R \frac{\partial l}{\partial \boldsymbol{A}} \boldsymbol{P}_C^T \cdot \boldsymbol{P}_C \boldsymbol{V}^T \boldsymbol{P}_R^T$$

$$= \boldsymbol{P}_R \frac{\partial l}{\partial \boldsymbol{A}} \boldsymbol{V}^T \boldsymbol{P}_R^T$$

$$= \boldsymbol{P}_R \frac{\partial l}{\partial \boldsymbol{S}} \boldsymbol{P}_R^T,$$

that is:

$$\frac{\partial l}{\partial \boldsymbol{S}_{(P)}} = \boldsymbol{P}_R \frac{\partial l}{\partial \boldsymbol{S}} \boldsymbol{P}_R^T. \tag{91}$$

Since $\boldsymbol{S}_{(P)}$ follows Eq. 73, we have the gradients for $\boldsymbol{Q}_{(P)}$ combining with Eq. 91:

$$
\begin{aligned}
\frac{\partial l}{\partial \boldsymbol{Q}_{(P)}} &= \frac{1}{\sqrt{d}}[(diag(\boldsymbol{S}_{(P)}) - \boldsymbol{S}_{(P)}^T \boldsymbol{S}_{(P)}) \frac{\partial l}{\partial \boldsymbol{S}_{(P)}}]\boldsymbol{K}_{(P)} \\
&= \frac{1}{\sqrt{d}}[(\boldsymbol{P}_R diag(\boldsymbol{S})\boldsymbol{P}_R^T - \boldsymbol{P}_R \boldsymbol{S}^T \boldsymbol{P}_R^T \cdot \boldsymbol{P}_R \boldsymbol{S} \boldsymbol{P}_R^T)\boldsymbol{P}_R \frac{\partial l}{\partial \boldsymbol{S}}\boldsymbol{P}_R^T]\boldsymbol{P}_R \boldsymbol{K} \boldsymbol{P}_C^T \\
&= \frac{1}{\sqrt{d}}[(\boldsymbol{P}_R diag(\boldsymbol{S})\boldsymbol{P}_R^T - \boldsymbol{P}_R \boldsymbol{S}^T \boldsymbol{S} \boldsymbol{P}_R^T)\boldsymbol{P}_R \frac{\partial l}{\partial \boldsymbol{S}}\boldsymbol{P}_R^T]\boldsymbol{P}_R \boldsymbol{K} \boldsymbol{P}_C^T \\
&= \frac{1}{\sqrt{d}}[\boldsymbol{P}_R (diag(\boldsymbol{S}) - \boldsymbol{S}^T \boldsymbol{S})\boldsymbol{P}_R^T \cdot \boldsymbol{P}_R \frac{\partial l}{\partial \boldsymbol{S}}\boldsymbol{P}_R^T]\boldsymbol{P}_R \boldsymbol{K} \boldsymbol{P}_C^T \\
&= \frac{1}{\sqrt{d}}[\boldsymbol{P}_R (diag(\boldsymbol{S}) - \boldsymbol{S}^T \boldsymbol{S}) \frac{\partial l}{\partial \boldsymbol{S}}\boldsymbol{P}_R^T]\boldsymbol{P}_R \boldsymbol{K} \boldsymbol{P}_C^T \\
&= \frac{1}{\sqrt{d}}\boldsymbol{P}_R (diag(\boldsymbol{S}) - \boldsymbol{S}^T \boldsymbol{S}) \frac{\partial l}{\partial \boldsymbol{S}}\boldsymbol{P}_R^T \cdot \boldsymbol{P}_R \boldsymbol{K} \boldsymbol{P}_C^T \\
&= \boldsymbol{P}_R \frac{1}{\sqrt{d}}(diag(\boldsymbol{S}) - \boldsymbol{S}^T \boldsymbol{S}) \frac{\partial l}{\partial \boldsymbol{S}}\boldsymbol{K} \boldsymbol{P}_C^T \\
&= \boldsymbol{P}_R \frac{\partial l}{\partial \boldsymbol{Q}} \boldsymbol{P}_C^T.
\end{aligned}
$$

By a similar derivation on $\boldsymbol{K}$ we obtain:

$$\frac{\partial l}{\partial \boldsymbol{K}_{(P)}} = \boldsymbol{P}_R \frac{\partial l}{\partial \boldsymbol{K}} \boldsymbol{P}_C^T. \tag{92}$$

Following a similar proof to the gradients of $\boldsymbol{W}_{1(P)}$ or $\boldsymbol{W}_{2(P)}$, we could easily derive:

$$\frac{\partial l}{\partial \boldsymbol{W}_{Q(P)}} = \boldsymbol{P}_C \frac{\partial l}{\partial \boldsymbol{W}_Q} \boldsymbol{P}_C^T, \tag{93}$$

$$\frac{\partial l}{\partial \boldsymbol{W}_{K(P)}} = \boldsymbol{P}_C \frac{\partial l}{\partial \boldsymbol{W}_K} \boldsymbol{P}_C^T. \tag{94}$$

And by Eq. 32, the gradient of $\boldsymbol{V}_{(P)}$ is

$$
\begin{aligned}
\frac{\partial l}{\partial \boldsymbol{V}_{(P)}} &= \boldsymbol{S}_{(P)}^T \frac{\partial l}{\partial \boldsymbol{A}_{(P)}} \\
&= \boldsymbol{P}_R \boldsymbol{S} \boldsymbol{P}_R^T \cdot \boldsymbol{P}_R \frac{\partial l}{\partial \boldsymbol{A}} \boldsymbol{P}_C^T \\
&= \boldsymbol{P}_R \frac{\partial l}{\partial \boldsymbol{V}} \boldsymbol{P}_C^T,
\end{aligned}
$$

thus we have

$$\frac{\partial l}{\partial \boldsymbol{V}_{(P)}} = \boldsymbol{P}_R \frac{\partial l}{\partial \boldsymbol{V}} \boldsymbol{P}_C^T, \tag{95}$$

$$\frac{\partial l}{\partial \boldsymbol{W}_{V(P)}} = \boldsymbol{P}_C \frac{\partial l}{\partial \boldsymbol{W}_V} \boldsymbol{P}_C^T. \tag{96}$$

So far, we have proved the rule for the gradient of weight matrices:

$$\frac{\partial l}{\partial \boldsymbol{W}_{i(P)}} = \boldsymbol{P}_C \frac{\partial l}{\partial \boldsymbol{W}_i} \boldsymbol{P}_C^T, \ \ i \in \{1, 2, Q, K, V\}. \tag{97}$$

$\boldsymbol{W}_{i(P)}$ are the weights of $Enc_{(P)}$ while $\boldsymbol{W}_i$ are the weights of $Enc$. With some induction, we can reach the conclusion that if a Transformer encoder blocks is randomly initialized and trained with $\boldsymbol{Z}_{(P)}$, it would eventually learn to become $Enc_{(P)}$, which is associated with $Enc$ by Eq. 97. $\qquad \square$

### B.6 PROOFS ON PARAMETERS OF THE EDGE

We show in this section that the parameters of the edge, including the weights associating position embeddings, are the same despite the shuffling method is used or not.

**Theorem 5.** *The parameters on the edge trained with or without row-column shuffling are the same.*

*Proof.* We denote the embedded feature in the naive SL and in the our shuffling scheme as $\boldsymbol{Z}_0, \boldsymbol{Z}_{0(P)}$, respectively, and the feature to be sent to the cloud in the two schemes as $\boldsymbol{Z}, \boldsymbol{Z}_{(P)}$. In naive split learning, $\boldsymbol{Z}_0 = \boldsymbol{Z}$ and $\frac{\partial l}{\partial \boldsymbol{Z}_0} = \frac{\partial l}{\partial \boldsymbol{Z}}$. In our scheme we have:

$$\boldsymbol{Z}_{(P)} = \boldsymbol{P}_R \boldsymbol{Z}_{0(P)} \boldsymbol{P}_C^T. \tag{98}$$

To prove the claim is to prove:

$$\frac{\partial l}{\partial \boldsymbol{Z}_{0(P)}} = \frac{\partial l}{\partial \boldsymbol{Z}_0}. \tag{99}$$

It is clear that

$$\begin{aligned}
\mathrm{d}l &= tr\left(\frac{\partial l}{\partial \boldsymbol{Z}_{(P)}}^T \mathrm{d}\boldsymbol{Z}_{(P)}\right) \\
&= tr\left(\boldsymbol{P}_C \frac{\partial l}{\partial \boldsymbol{Z}}^T \boldsymbol{P}_R^T \boldsymbol{P}_R \mathrm{d}\boldsymbol{Z}_{0(P)} \boldsymbol{P}_C^T\right) \\
&= tr\left(\boldsymbol{P}_C^T \boldsymbol{P}_C \frac{\partial l}{\partial \boldsymbol{Z}}^T \mathrm{d}\boldsymbol{Z}_{0(P)}\right) \\
&= tr\left(\frac{\partial l}{\partial \boldsymbol{Z}}^T \mathrm{d}\boldsymbol{Z}_{0(P)}\right),
\end{aligned}$$

Hence,

$$\frac{\partial l}{\partial \boldsymbol{Z}_{0(P)}} = \frac{\partial l}{\partial \boldsymbol{Z}} = \frac{\partial l}{\partial \boldsymbol{Z}_0}. \tag{100}$$

The second equality holds by a similar argument to Eq. 95. The equivalence between the edge weights in the two schemes is accomplished by Eq. 100 and the fact that their forward procedure is exactly the same. □

## C SUPPLEMENTARY EXPERIMENTS

### C.1 VERIFYING THE PROPERTIES ON BIAS AND LAYER NORM

We do not mention bias and $\gamma$, the weight in layer normalization, in the proofs in Appendix B. Here is some intuition about why they are encrypted in the way shown in Eq. 10:

$$b_{(P)} = b\boldsymbol{P}_C^{-1}, \ \ \gamma_{(P)} = \gamma \boldsymbol{P}_C^{-1}.$$

Both bias and $\gamma$ are 1-D vector, and each element only interrelates with the corresponding column of $\boldsymbol{Z}$ in both forward and backward propagation. So if the columns are permuted, bias and $\gamma$ should be permuted in the same way. During the experiments we find that the encryption/decryption of bias and $\gamma$ hardly affect the model performance on Cifar10. But on CelebA, if most weight matrices are encrypted by $M_{\boldsymbol{W}}$ while bias and $\gamma$ are not permuted, the accuracy would be greatly affected, which is merely 79.056%. But if the encryption is strictly performed as Eq. 10, the relative accuracy difference between a normally trained model (tested with $\boldsymbol{Z}$) and an encrypted model (tested with $\boldsymbol{Z}\boldsymbol{P}_C^T$) is merely 0.00013% (91.50991% and 91.50979% respectively).

### C.2 VERIFYING ON ORDER-DEPENDENT TASKS

We consider the classification task is weakly order-dependent, i.e., the task may rely little on the order of the patches of an input image. To verify the feasibility of our method on strictly order-dependent tasks, we designed a simple task which strongly depends on the input order. We label

the shuffled images in CelebA as 1 and the original ones as 0, and train a ViT-Base on the them from scratch to distinguish whether the images are shuffled. Within one epoch, the accuracy of ViT-Base reaches around 97%, showing the model and the task are strictly order-dependent. And we conduct similar property-verifying experiments in Sec. 6.1, and the conclusion does not change: RS is transparent to the encoder, and encrypted models can only process encrypted data. It demonstrates the shuffled Transformer also works on tasks strongly associated with the input patch order.

### C.3 VERIFYING ON NLP DATA

We implement a small Bert model and use a pre-trained model to fine-tune it for natural language inference on the SNLI dataset[2]. The small Bert has 2 layers and the input $Z$ is of shape $(batch\_size, 128, 256)$. Due to the non-square MLP (with a hidden dimension of 512), the column shuffle is not suitable for this pre-trained model. Noting that the position embedding is not removed from the model. With or without the row shuffle method, the small Bert achieves the similar accuracies: 76.4% and 76.5% respectively. The code is provided in supplementary materials.

### C.4 VERIFYING ON TABULAR DATA

We use the DIFM (Lu et al., 2021) model and Criteo dataset to verify the proved properties on tabular data. The vector-wise part of DIFM model is a Transformer encoder block. We shuffle the rows of the vector-wise part of the input and unshuffle its output. The model achieves the same AUC, 0.777, with or without the row shuffle method.

In summary, our method works on CV, NLP and tabular data, despite the specific tasks. This is attributed to the modeling where $Z$ is a general matrix, and the permutation invariance property holds with no strings attached. Speaking of invariance, a negligible $10^{-7}$ error occurs per element in the permutation due to float calculation error.

### C.5 ATTACK RESULTS TO CIFAR10

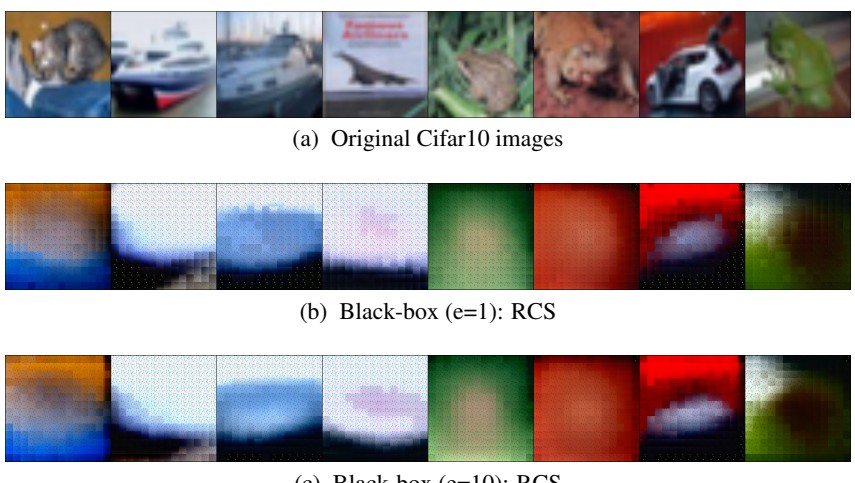

(a) Original Cifar10 images

(b) Black-box (e=1): RCS

(c) Black-box (e=10): RCS

Figure 8: Results of the black-box attack to the private inference data of Cifar10.

We provide additional experimental results on Cifar10 in Fig. 8 and Tab. 6. Black-box attacks described in Sec. 3 is launched. Due to the sparse data distribution, the defence on Cifar10 is more successful than on CelebA. RCS almost eliminates the sketch of the object in each reconstructed image in Fig. 8.

Table 6: Results of the black-box attack to the private training and testing data of Cifar10. ↓ means desirable direction.

|  | SSIM↓ | PSNR↓ | F-SIM↓ |
|---|---|---|---|
| unprotected SL | 0.360 | 2.761 | 0.644 |
| Our RS (inference) | 0.075 | 0.987 | 0.464 |
| Our RCS (inference) | 0.094 | 0.984 | 0.421 |
| Our RS (training) | 0.275 | 10.812 | 0.390 |
| Our RCS (training) | 0.264 | 10.460 | 0.336 |



(a) Normally Learned     (b) Before Shuffle     (c) After Shuffle

Figure 9: Cosine similarity between every two patches of position embeddings learned in different ways.

## C.6 EXPERIMENTS ON POSITION EMBEDDINGS

To further verify that our method has little impact to network parameters on the edge as proved in Appendix B.6, we visualized in Fig. 9 the cosine similarity of position embeddings learned at different places of the model on the CelebA classification tasks. We found that if the position embedding is placed ahead of shuffling, the cosine similarity shares a similar state to that without shuffling, as shown in Fig. 9, resembling a human face. Hence our shuffling method almost does not vary the weights on the edge.

To verify the impact of the removal of position embeddings in training, we train the Transformer from scratch and on pre-trained ones, both on large and small datasets. Except for training from scratch on Cifar10, all model accuracies maintain at a similar level to that with position embeddings. In the exceptional case, the accuracy drops by $\sim 10\%$ ($\sim 80\%$ with position embeddings and $\sim 70\%$ otherwise). We consider it mainly due to the poor performance of the natural Transformer on small datasets. After all, Transformer works best with pre-training on a large dataset.

## C.7 ABLATION STUDY OF PATCH SIZE

Table 7: The privacy, utility and efficiency when selecting different patch size on cifar10. The width and height of input image are both 32 pixels. ↓ means desirable direction.

| patch size | num. | Utility Accuracy↑ | Privacy JigsawGAN Acc↓ | Efficiency infer time (2k images) |
|---|---|---|---|---|
| $2 \times 2$ | 256 | 82.64% | <0.1% | 33s659ms |
| $4 \times 4$ | 64 | 79.67% | 1.16% | 7s751ms |
| $8 \times 8$ | 16 | 77.25% | 60.03% | 2s186ms |
| $16 \times 16$ | 4 | 66.51% | 83.52% | 694ms |

The selection of different patch size have significant influences on the privacy, accuracy and efficiency. For fixed input image size, increasing the patch size will decrease the total number of patches which are the basic unit of shuffling. As shown in Table 7, increasing the patch size obvi-

---
[2]https://nlp.stanford.edu/projects/snli/

ously degrades the accuracy, since the performance of ViT is affected by decreased patch numbers. Revealed by the results of JigsawGAN (Li et al., 2021) (using GAN model to reorganize the shuffled patches, aka. Jigsaw problem) and black-box attack, increasing the patch size also has harmful impact on the privacy, which indicates that larger size of patches carry more information in one basic unit of shuffling and make it easier for attacker to solve the relationships between patches. However, larger patch size and fewer patches significantly shorten the computation cost, as the time complexity of Attention block is quadratically proportional to the number of patches.

