# OpenReview forum: "Shuffled Transformers for Blind Training"
_ICLR.cc/2023/Conference — Submitted to ICLR 2023_

### Official Review · Reviewer_N7YY · 2022-10-23

**Confidence:** 3
**Correctness:** 2
**Technical Novelty And Significance:** 2
**Empirical Novelty And Significance:** 2
**Recommendation:** 3

**Clarity, Quality, Novelty And Reproducibility:**

-) Quality of the paper is difficult to evaluate based on the limited clarity of the method.
+) The paper is original of its own.

**Strength And Weaknesses:**

Pros:
+) The paper proposes an interesting topic of performance privacy protection on shuffling input features in the Transformer structure;
+) The paper provides theoretical deduction on shuffled transformer;

Cons:
-) The motivation is not clearly elaborated. Examples of privacy leakage of transformers should be provided to validate the importance of designing a shuffled transformer;
-) After shuffling, is the positional encoding not working on such Transformer structure?
-) I'm curious about the performance degration with the proposed method compared with naive transformer structure. More analysis should be given.
-) The paper is very hard to follow. Authors should provide a more clear elaboration of the proposed method.
-) Writing could be improved.

**Summary Of The Paper:**

This paper proposes a shuffled transformer structure for privacy preserving. The networks are split into two parts, where the first part encodes the input data and the the input to the second part network is shuffled features without knowing the original data. The experiments have shown superior performance against diverse adversarial attacks.

**Summary Of The Review:**

The paper proposes a shuffled transformer for privacy protection. The idea is interesting and original, and the authors have provided a detailed theoretical deduction of the inference stage. However, the motivation seems vague to me and it is very hard to follow. I have doubts of whether such a shuffling on the feature space will affect the position encoding, and thus I'm not convinced by the qualitative and quantitative results.

---

> ### Author Response · Authors · 2022-11-09
> **Response to Reviewer N7YY**
>
> We thank the reviewer for the valuable comments and have modified the structure of our paper to improve clarity.
>
> "The motivation is not clearly elaborated. Examples of privacy leakage of transformers should be provided to validate the importance of designing a shuffled transformer"
> - We show evidence of privacy leakage in split learning in Sec. 2.
> - Particularly for Transformers, we offer its privacy leakage by the experiments on unprotected split learning, in Tab. 4 and Fig. 4.
>
> "After shuffling, is the positional encoding not working on such Transformer structure? "
> - The position encoding still works. If the position encoding is placed ahead of the shuffle, it learns as normal (see Appendix C.6). If it is placed after the shuffle (just to showcase the effect, not part of our scheme), the position encoding learns random things. The position encoding still works because the model weights on the edge do not change with shuffling, as we show theoretically in Appendix B.6.
>
> "I'm curious about the performance degration ... More analysis should be given."
> - In fact, all results on unprotected split learning of our paper are reported on the naive transformer without modification. As Tab.4 shows, the modification to the Transformer structure incurs approximately 0.1-0.4 percentage of accuracy loss.

---

### Official Review · Reviewer_pVBj · 2022-10-25

**Confidence:** 2
**Correctness:** 4
**Technical Novelty And Significance:** 3
**Empirical Novelty And Significance:** 3
**Recommendation:** 5

**Clarity, Quality, Novelty And Reproducibility:**

The writing of this paper is understandable.
This is no code attached for reproducibility.

**Details Of Ethics Concerns:**

No ethics concerns.

**Strength And Weaknesses:**

strength:

This work focuses on a valuable problem – data and model privacy preservation. The authors give a clear problem statement about the concept of blind training. Moreover, experiment results prove the effectiveness of the proposed method.

weakness:

1. The authors state that “P_R can be chosen randomly for each Z, while P_C is randomly chosen once for the model”. How the row and column shuffling strategy is determined?

2. What are the differences between row shuffling in the paper with the original training data shuffling operations?

3. Since only one type of column shuffling is adopted for each model, will the performance be affected by the highly correlated features? If two dimensions have large information overlap, swapping them seems have a small impact on the performance. Did the authors consider such a situation? Such a situation is beneficial or harmful for the proposed method?

4. What does ⊥ in Equ. (7) refer to? Perpendicular?





**Summary Of The Paper:**

This paper introduces blind training to preserve the data and model privacy via shuffled Transformers. An intriguing finding is proposed that, inputs and the model weights of the Transformer encoder blocks, the backbone of Transformer, can be shuffled without degrading the model performance.

**Summary Of The Review:**

This paper proposed a blind training method to realize privacy-preserving split learning, where the cloud trains over unknown data and model for the edge. Theoretical proofs, property verification, and real-world performance-resisting attacks are provided. The method successfully
defends black-box, and white-box attacks without degrading accuracy and efficiency. However, there are some concerns/ weaknesses need to be clarified.

---

> ### Author Response · Authors · 2022-11-09
> **Response to Reviewer pVBj**
>
> We thank the reviewer for the valuable comments. Our replies are as follows.
>
> "How the row and column shuffling strategy is determined?"
> - We elaborate shuffling strategy at the beginning of Sec. 4 and provide the pseudo-code in Appendix A in the revision.
>
> "What are the differences between row shuffling in the paper with the original training data shuffling operations?"
> - The training data shuffling operations permute the order of input data. But our shuffle works inside an input instance, or its embedded feature, to be more precise. Letting the input to encoder be of shape (batch-size, number-of-patch, resolution), the training data shuffling operations apply to the first dimension, while our row shuffle works on the second dimension, and column shuffle on the third dimension. More demonstrations of our shuffling method are presented in Appendix A in the revision.
>
> "..., will the performance be affected by the highly correlated features? If ... Did the authors consider such a situation? Such a situation is beneficial or harmful for the proposed method?"
> - The `column' in column-shuffling refers to the internal element order within one patch embedding. Hence features corresponding to different pixels within the same patch do have large information overlap, which does not affect the accuracy according to the analysis, but indeed impacts privacy.
> - Since the `row' --- meaning different patches are randomly ordered for different inputs, the features corresponding to different patches have little correlation and thus when the patch dimension is small and the number of patches is large, we could ignore the impact of column correlations.
> - We designed our privacy-preserving framework from the inherent properties of Transformers, not the other way around. Hence it may not be perfect.
>
> "What does ⊥ in Equ. (7) refer to?"
> - Sorry for the confusion. We replace it with "Invalid Result."

---

### Official Review · Reviewer_kXfb · 2022-10-26

**Confidence:** 3
**Correctness:** 4
**Technical Novelty And Significance:** 3
**Empirical Novelty And Significance:** 3
**Recommendation:** 8

**Clarity, Quality, Novelty And Reproducibility:**

**Clarity.** Fair: The writing, figures and tables are clear, but the english is poor.

**Quality.** Good: The paper appears to be technically sound. The proofs, appear to be correct, but I have not carefully checked the details. The experimental evaluation, is adequate, and the results convincingly support the main claims.

**Novelty.** Good: The paper makes non-trivial advances over the current state-of-the-art.

**Reproducibility.** Excellent: key resources (e.g., proofs, code, data) are available and key details (e.g., proof sketches, experimental setup) are comprehensively described such that competent researchers will be able to easily reproduce the main results.

**Strength And Weaknesses:**

# Strengths

* Novel algorithm. The main idea of using the shuffle invariance property of transformers to obfuscate the client data and model weights during split learning is ingenious, novel, and well motivated.

* Novel theoretical results. Analysis of the shuffling invariance property of transformers is novel and provides a strong theoretical foundation for the proposed framework. Privacy definition for shuffling mechanisms is also interesting and novel, albeit a bit tangental.

* Good results. The proposed method achieves a favorable privacy-utility trade-off compared to relevant baselines. This is demonstrated clearly with experiments on the CelebA and CIFAR10 datasets.

# Weaknesses

* Poor writing. The paper would benefit greatly from a review by a native english editor.

**Summary Of The Paper:**

The paper proposes *blind learning*, a novel split learning framework for training transformers. In this framework, the patch embedding, MLP, and loss layers reside at the client, and the transformer blocks reside at the server. During training and inference, the client obfuscates its data and model weights by shuffling the patch embeddings before sending them to the server, and then unshuffles the outputs of the transformer blocks sent back by the server. Crucially, since transformer blocks are invariant to random shuffling of the input data, the proposed shuffling by the client does not degrade model training or inference. Theoretical analysis and experimental evaluations demonstrating the transformer invariance to shuffling are provided. A privacy analysis of the proposed framework is also provided. A novel privacy definition for shuffling mechanisms is also provided. Lastly, the proposed framework is compared against various baselines with respect to classification accuracy for a target task and susceptibility to black-box and white-box reconstruction attacks.

**Summary Of The Review:**

Novel framework and theoretical analysis. Strong results.

---

> ### Author Response · Authors · 2022-11-09
> **Response to Reviewer kXfb**
>
> We appreciate the reviewer for the valuable comments. We have revised the paper structure for better understanding, e.g., introducing the properties of Transformers in front of the solution.

---

### Official Review · Reviewer_jfZZ · 2022-10-31

**Confidence:** 2
**Correctness:** 3
**Technical Novelty And Significance:** 2
**Empirical Novelty And Significance:** 2
**Recommendation:** 5

**Clarity, Quality, Novelty And Reproducibility:**

The clarity can be improved by giving the definition of the "rows" and "columns" in a more obvious place.

**Strength And Weaknesses:**

Strength: The proposed approach can improve the privacy of edge users.

Weaknesses:

1. The experiments are limited to vision Transformers, but the title seems to over claim it to any Transformer. For languages, it should be hard to train without position embeddings, as the orders of the tokens are important. I would suggest the authors to either modify the title or show some empirical evidence that the method also works for language models.

2. I am quite confused with the notion of row and column shuffling. How are the matrix multiplications defined (Wx or xW)? At the beginning of Section 3, each X is just a vector. Do you shuffle on the patch level or pixel level? If you only change the order of the patches, isn't it trivial that the output won't change when position embedding is not used?


**Summary Of The Paper:**

This paper proposes a shuffling based method to improve the privacy and enable a split learning paradigm, where the cloud server only requires shuffled data from the edge.

**Summary Of The Review:**

I feel the paper lacks clarity in both the scope and the technical details. The proposed method might just be a trivial shuffling of the image patches without position encodings, and therefore lack technical novelty.

---

> ### Author Response · Authors · 2022-11-09
> **Response to Reviewer jfZZ**
>
> We thank the reviewer for the valuable comments. Our replies are as follows.
>
> “The experiments are limited to vision Transformers ... either modify the title or show some empirical evidence ..."
> - Our work can be applied to any Transformer. Our method works for NLP and tabular data as well, as long as the model structure meets the requirement. and the experimental results are provided in Appendix C.
>
> " ... quite confused with the notion of row and column shuffling ... isn't it trivial that... when position embedding is not used?"
> - We apologize for the confusion on the notations and have revised them accordingly. Row/column shuffle is defined in the 1st paragraph of Sec.4, and further examples are offered in Appendix A.
> - We made mistakes on the shape of $X$ and the paper is about patch, not pixel shuffling. The definition $Z$ is given in the 2nd paragraph of Sec. 5.1.
> - We have revised the final paragraph of Sec. 6.1 to include the discussion on positional embeddings. Our work showed that shuffling equivalence/invariance holds despite whether positional embeddings exist or not.

---

### Author Response · Authors · 2022-11-09
**Summary of Revision**


Newly included or heavily modified sections: Sec. 4, Sec. 6.1, Appendix A, B.6, C.3, C.4.

- We re-organized the paper structure by moving the properties analysis in front of the solution framework.
- We resolved the confusion on shuffling by specifying definitions in Sec. 4 and examples in Appendix A.
- We clarified the role of position embedding in the last paragraph of Sec 6.1. Theoretical derivation is given in Appendix B.6. Empirical evidence is provided in Appendix C.6.
- We provided experimental results on NLP (with Bert and SNLI dataset) and tabular data (with DIFM and Criteo dataset) and showed that our method not only works for CV, but also for other applications. Please find them in Appendix C.3 and C.4.
- Code is provided in the supplementary.
- A colored version of our revision is provided in the supplementary material where revised contents are in blue.

---

### Decision · Program_Chairs · 2023-01-20

**Decision:**

Reject

**Justification For Why Not Higher Score:**

Poorly written.

**Justification For Why Not Lower Score:**

N/A

**Metareview: Summary, Strengths And Weaknesses:**

This paper proposes a shuffling-based method to improve privacy and enable a split learning paradigm, where the cloud server only requires shuffled data from the edge. The proposed approach can improve the privacy of edge users. The idea is novel both the algorithm and the theory. The results are good.


One of the main limitations of this work is that it only works for vision transformers and the article overclaims. The notion is confusing. Poor writing. The paper would benefit greatly from a review by a native English editor. The motivation is not clearly elaborated. Examples of privacy leakage of transformers should be provided to validate the importance of designing a shuffled transformer.

The authors have done a great job addressing many of the reviewer's comments:
 - Newly included or heavily modified sections: Sec. 4, Sec. 6.1, Appendix A, B.6, C.3, C.4.
- They reorganized the paper structure by moving the properties analysis in front of the solution framework.
- They resolved the confusion on shuffling by specifying definitions in Sec. 4 and examples in Appendix A.
- They clarified the role of position embedding in the last paragraph of Sec 6.1. The theoretical derivation is given in Appendix B.6. Empirical evidence is provided in Appendix C.6.
- They provided experimental results on NLP (with Bert and SNLI dataset) and tabular data (with DIFM and Criteo dataset) and showed that our method not only works for CV, but also for other applications. Please find them in Appendix C.3 and C.4.
- Code is provided in the supplementary.

However, the paper still does not pass the bar.